# Generalization Bounds with Logarithmic Negative-Sample Dependence for Adversarial Contrastive Learning

**Naghmeh Ghanooni**                                                    *ghanooni@cs.uni-kl.de*
*Department of Computer Science*
*RPTU, Kaiserslautern, Germany*

**Waleed Mustafa**                                                      *mustafa@cs.uni-kl.de*
*Department of Computer Science*
*RPTU, Kaiserslautern, Germany*

**Yunwen Lei**                                                          *leiyw@hku.hk*
*Department of Mathematics*
*University of Hong Kong, Hong Kong, China*

**Anthony Widjaja Lin**                                                 *lin@cs.uni-kl.de*
*MPI-SWS*
*RPTU, Kaiserslautern, Germany*

**Marius Kloft**                                                        *kloft@cs.uni-kl.de*
*Department of Computer Science*
*RPTU, Kaiserslautern, Germany*

**Reviewed on OpenReview:** *https://openreview.net/forum?id=OaVi1yjdEc*

## Abstract

Contrastive learning has emerged as a powerful unsupervised learning technique for extracting meaningful representations from unlabeled data by pulling similar data points closer in the representation space and pushing dissimilar ones apart. However, its vulnerability to adversarial attacks remains a critical challenge. To address this, adversarial contrastive learning — incorporating adversarial training into contrastive loss — has emerged as a promising approach to achieving robust representations that can withstand various adversarial attacks. While empirical evidence highlights its effectiveness, a comprehensive theoretical framework has been lacking. In this paper, we fill this gap by introducing generalization bounds for adversarial contrastive learning, offering key theoretical insights. Leveraging the Lipschitz continuity of loss functions, we derive generalization bounds that scale logarithmically with the number of negative samples, $K$, and apply to both linear and non-linear representations, including those obtained from deep neural networks (DNNs). Our theoretical results are supported by experiments on real-world datasets.

## 1 Introduction

Learning meaningful representations from unlabeled data plays a crucial role in enhancing the performance of machine learning models. Representation learning has shown great success in fields such as computer vision (Chen et al., 2020b; He et al., 2020; Caron et al., 2020) and natural language processing (Brown et al., 2020; Gao et al., 2021; Radford et al., 2021). Among various representation learning techniques, self-supervised contrastive learning (CL), popularized by the SimCLR framework (Chen et al., 2020b), stands out. The core idea behind contrastive learning is to bring similar pairs $(x, x^+)$ closer together in the embedding space while pushing apart negative samples from $x$, (denoted as $(x, x_1^-, \cdots, x_K^-)$). These learned representations can then be leveraged for downstream tasks, such as a classification task, whether

supervised or unsupervised (Chen et al., 2020b; He et al., 2020; Khosla et al., 2020). Notably, extensive research in contrastive learning has revealed that a sufficient number of negative samples is essential for achieving high-quality representations (Chen et al., 2020b; Khosla et al., 2020; Henaff, 2020; Tian et al., 2020).

Despite significant progress in representation learning, these representations remain susceptible to adversarial examples (Szegedy et al., 2013; Biggio et al., 2013), which are subtly perturbed samples carefully crafted to manipulate a model's predictions. Specifically, adversarial attacks aim to maximize the model's loss by slightly perturbing input samples. To mitigate this vulnerability, researchers have proposed adversarial training (Chen et al., 2020a; Tramer & Boneh, 2019). This technique employs a min-max optimization approach, where the model simultaneously minimizes its loss while facing maximally perturbed examples. By doing so, adversarial training enhances the robustness of the learned representations against adversarial attacks. Adversarial contrastive learning (ACL) emerges from applying adversarial training to contrastive learning. In this paradigm, adversarial training enhances the robustness of representations learned from unlabeled data during unsupervised training. Empirical evidence supports the effectiveness of ACL in improving the quality of these robust representations (Kim et al., 2020; Ho & Nvasconcelos, 2020; Jiang et al., 2020). Despite its empirical success, the theoretical foundations of ACL remain somewhat limited.

Recent work by Zou & Liu (2023) leverages Rademacher complexity to show the connection between unsupervised contrastive learning and the downstream classification task and claimed that the average adversarial risk of downstream tasks can be upper bounded by the adversarial unsupervised risk of the upstream task. Specifically, they derive a surrogate upper bound for the adversarial risk by analyzing the average supervised risk. In the case of a single negative sample, they bound the adversarial supervised risk using the surrogate unsupervised risk. For multiple negative samples, they introduce an average adversarial supervised risk, which is similarly bounded by the surrogate unsupervised risk. This results in a bound that scales as $\mathcal{O}(K)$, where $K$ represents the number of negative samples, leading to a linear dependence on $K$. However, their approach does not fully exploit the coupling between negative samples, resulting in suboptimal bounds. As a result, their approach is not well-suited for scenarios involving a large number of negative samples, which is essential for achieving optimal generalization performance (Chen et al., 2020b; Tian et al., 2020; Henaff, 2020; Khosla et al., 2020). In contrast, our work extends this analysis to the general case with a large number of negative samples, aiming to improve the dependency on the sample size and provide more effective bounds.

In this paper, we present the following contributions:

- We apply the $\ell_\infty$-Lipschitz property of loss functions to derive generalization error bounds for ACL. These bounds incorporate the covering number of feature classes and show improved dependency on the number of negative examples, resulting in tighter bounds compared to existing literature by Zou & Liu (2023).

- Our general results are applied to two specific scenarios of unsupervised representation learning: learning linear features and learning non-linear features via DNNs. In both cases, the bounds show a logarithmic dependence on the number of negative samples.

The remainder of the paper is organized as follows: Section 2 reviews related work and state-of-the-art approaches. In Section 3, we define the problem and set up our framework. Our main theorem on the generalization error bound for ACL is presented in Section 4. Section 5 applies this result to both linear and nonlinear feature representations, demonstrating the corresponding generalization bounds. Empirical evaluations are provided in Section 6. All proofs for the lemmas and theorems can be found in Section 7 and the appendix. Finally, Section 8 concludes the paper.

## 2 Related work

**Contrastive Learning** Our work is primarily related to the theoretical analysis of contrastive learning by Arora et al. (2019) and Lei et al. (2023). Arora et al. (2019) provided generalization bounds for contrastive learning by analyzing the Rademacher complexity of the representation function class and examining the performance of linear classifiers trained on the learned representations. They showed that the classification

error of a mean classifier is bounded by the unsupervised errors of learned representation functions, indicating that downstream tasks such as classification tasks benefit from representations with low unsupervised errors. However, their generalization bounds scale linearly with $K$, the number of negative samples, which becomes impractical when $K$ is large, as is often the case in CL. Motivated by this, Lei et al. (2023) improved the dependence on the number of negative samples. For $\ell_2$ Lipschitz loss, their bound is independent of $K$, and for $\ell_\infty$ Lipschitz loss, they achieved a reduction by a factor of $K$. Ji et al. (2023) introduced a theoretical framework for contrastive learning under the linear representation setting, providing a detailed analysis of the feature learning performance in the spiked covariance model. Their work theoretically justified why contrastive learning can remove more noise compared to autoencoders and GANs by constructing contrastive samples via augmentations. In a PAC-Bayesian setting, Nozawa et al. (2020) derived PAC-Bayesian generalization bounds on the posterior distribution of representation functions. Another key challenge in CL is the random selection of negative samples, which can result in some negatives sharing the same label as the anchor point. This introduces bias into the CL loss function and can potentially reduce performance in practice. Chuang et al. (2020) addressed this by deriving an approximation for the unbiased contrastive loss and establishing generalization bounds for downstream tasks.

**Adversarial Robustness**   Since Szegedy et al. (2013) first revealed the vulnerability of neural networks to small input perturbations (Jiang et al., 2020; Kim et al., 2020), numerous studies have focused on establishing generalization bounds for adversarial learning, primarily in the supervised setting. Montasser et al. (2019) explored the PAC learnability of adversarial robust learning, and Xu & Liu (2022) extended these findings to the multi-class classification problem. Additionally, several works have used Rademacher complexity to analyze adversarial learning under $\ell_p$-norm additive perturbation attacks (Yin et al., 2019; Awasthi et al., 2020; Khim & Loh, 2018; Xiao et al., 2022; Mustafa et al., 2022). Yin et al. (2019) derived Rademacher complexity-based bounds for linear models and single-layer neural networks using a surrogate loss, while Awasthi et al. (2020) introduced bound based on the direct loss for linear models and two-layer neural networks. Expanding on this approach, Mustafa et al. (2022) developed bounds for a broader range of attacks, directly applied to the loss function, and showed that their results grow at a rate of $\mathcal{O}(\log C)$, where $C$ is the number of label classes. In contrast, Khim & Loh (2018) proposed a tree-transform method to propagate adversarial noise through the network, leading to a bound that scales exponentially with the number of classes, $\mathcal{O}(C)$.

**Adversarial Contrastive Learning**   Recent studies have increasingly applied adversarial training to contrastive learning loss to improve model robustness (Kim et al., 2020; Ho & Nvasconcelos, 2020; Jiang et al., 2020). However, the theoretical foundations of ACL remain underexplored. Zou & Liu (2023) leveraged Rademacher complexity to show that the average adversarial risk in downstream tasks can be bounded by the adversarial unsupervised risk of the upstream task. Specifically, they derived a surrogate upper bound for the adversarial risk by analyzing the average supervised risk. For the case of a single negative sample, they provided bounds on the adversarial supervised risk using the surrogate unsupervised risk and extended their approach to multiple negative samples by introducing an average adversarial supervised risk, also bounded by the unsupervised risk. However, their bound scales linearly as $\mathcal{O}(K)$ with the number of negative samples $K$. A concurrent work studied adversarial contrastive learning by using structural result on infinity-norm covering numbers (Wen et al.).

## 3   Problem Formulation

### 3.1   Contrastive Representation Learning

In the contrastive learning setting, we aim to learn representations by contrasting similar and dissimilar data points. Let $\mathcal{X}$ be an input space (e.g., a set of input images). Given an anchor sample $x$, we use a positive sample $x^+$, which is drawn from a distribution of similar data $\mathcal{D}_{\text{sim}}$, and multiple negative samples $x_1^-, x_2^-, \cdots, x_K^-$, which are drawn from a negative distribution $\mathcal{D}_{\text{neg}}$. The goal is to learn a representation where the anchor and positive pair are pulled closer together, while the anchor and negative samples are pushed apart in the feature space. In this setup, a single positive sample is paired with multiple negative samples, creating an inherent asymmetry that is standard in both theoretical (Arora et al., 2019; Lei et al.,

2023) and empirical contrastive learning (Chen et al., 2020b; Khosla et al., 2020). This approach simplifies implementation and analysis while reducing computational complexity. Adding more positive samples does not always yield proportional performance improvements and can increase the risk of overfitting by reducing contrast between similar and dissimilar examples. Conversely, multiple negative samples are essential for enhancing feature representations, as they offer diverse examples for the model to distinguish from the positive pair. We follow the framework of Arora et al. (2019) to define the distribution of $\mathcal{D}_{\text{sim}}$ and $\mathcal{D}_{\text{neg}}$. The distributions $\mathcal{D}_{\text{sim}}$ and $\mathcal{D}_{\text{neg}}$ are generally characterized through a set of latent classes $\mathcal{C}$ and an associated probability distribution $\rho$ over these classes. For each latent class $c \in \mathcal{C}$, let $\mathcal{D}_c$ be the conditional distribution of the inputs given the latent class $c$. $\mathcal{D}_{\text{sim}}$ and $\mathcal{D}_{\text{neg}}$ are defined as:

$$\mathcal{D}_{\text{sim}}(x, x^+) = \mathbb{E}_{c \sim \rho}[\mathcal{D}_c(x)\mathcal{D}_c(x^+)],$$

$$\mathcal{D}_{\text{neg}}(x^-) = \mathbb{E}_{c \sim \rho}[\mathcal{D}_c(x^-)].$$

That is, $\mathcal{D}_{\text{sim}}(x, x^+)$ measures the probability of drawing $x$ and $x^+$ from the same class $c \sim \rho$, which means that $x$ and $x^+$ are conditionally dependent, given $c$, while $\mathcal{D}_{\text{neg}}(x^-)$ measures the probability of drawing $x^-$ that is independent of $x$ and $x^+$, coming from other latent classes. The objective of CL is to select a feature map $f : \mathcal{X} \to \mathbb{R}^d$ from a class of representation functions $\mathcal{F} = \{f : \|f(\cdot)\|_1 \leq R\}$, for some $R > 0$, where $\|\cdot\|_1$ denotes the $\ell_1$-norm, and $d \in \mathbb{N}$ represents the dimensionality of the feature space. This is achieved using the training set

$$S = \{(x_1, x_1^+, x_{11}^-, \cdots, x_{1K}^-), (x_2, x_2^+, x_{21}^-, \cdots, x_{2K}^-), \cdots, (x_n, x_n^+, x_{n1}^-, \cdots, x_{nK}^-)\},$$

where $(x_j, x_j^+) \sim \mathcal{D}_{\text{sim}}$ and $(x_{j1}^-, \cdots, x_{jK}^-) \sim \mathcal{D}_{\text{neg}}^K$, with $j \in [n] := \{1, \cdots, n\}$ and $K$ indicating the number of negative samples. However, the specific distributions $\mathcal{D}_{\text{sim}}$ and $\mathcal{D}_{\text{neg}}$ are abstracted out once we are dealing with a fixed dataset $S$. The quality of the representation $f$ is evaluated using the loss $\ell\left(\{f(x)^T(f(x^+) - f(x_k^-)\}_{k=1}^K\right)$, where $\ell : \mathbb{R}^K \to [0, B]$ is some loss function and $f(x)^T$ is the transpose of $f(x)$. The population and empirical risks are then defined as follows.

**Definition 3.1** (Upstream unsupervised risk). The population unsupervised risk is defined as:

$$L_{\text{un}}(f) = \mathbb{E}[\ell(\{f(x)^T(f(x^+) - f(x_k^-))\}_{k=1}^K)],$$

and the empirical unsupervised risk on $S$ is defined as:

$$\widehat{L}_{\text{un}}(f) = \frac{1}{n}\sum_{i=1}^n \ell(\{f(x_i)^T(f(x_i^+) - f(x_{ik}^-))\}_{k=1}^K)].$$

To find the representations $f$ in an unsupervised manner, we employ an unsupervised loss function $\ell : \mathbb{R}^K \mapsto \mathbb{R}_+$ which can be chosen to be a hinge loss. Given a vector $u$, the hinge loss is defined as:

$$\ell(u) = \max\{0, 1 + \max_{i \in [K]}\{-u_i\}\}.$$

## 3.2 Adversarial Contrastive Representation Learning

In this paper, we examine adversarial settings where an attacker employs a noise function $A : \mathcal{X} \times \mathcal{B} \to \mathcal{X}$, with $\mathcal{B}$ being a noise set, to subtly introduce noise $\delta \in \mathcal{B}$ to an input $x \in \mathcal{X}$ in order to maximize the loss. For example, in the $L_p$-additive attack, $A(x, \delta) = x + \delta$ and $\mathcal{B}$ is the $\ell_p$-ball $\{\delta : \|\delta\|_p \leq \beta\}$. The attacker's objective is to select $\delta^* \in \mathcal{B}$ that maximizes the loss:

$$\delta^* = \arg\max_{\delta \in \mathcal{B}} \ell(\{f(A(x, \delta))^T(f(x^+) - f(x_k^-))\}_{k=1}^k).$$

The adversarial and empirical risks are subsequently defined as follows.

**Definition 3.2.** (Unsupervised adversarial contrastive risk). The population unsupervised adversarial contrastive risk is defined as:

$$L_{\text{un}}^{\text{adv}}(f) = \mathbb{E}[\max_{\delta \in \mathcal{B}} \ell(\{f(A(x, \delta))^T(f(x^+) - f(x_k^-))\}_{k=1}^K)]$$

and the empirical unsupervised adversarial contrastive risk is defined as:

$$\hat{L}_{\text{un}}^{\text{adv}}(f) = \frac{1}{n} \sum_{i=1}^{n} \max_{\delta \in \mathcal{B}} \ell(\{f(A(x_i, \delta))^T (f(x_i^+) - f(x_{ik}^-))\}_{k=1}^{K})].$$

In our definition of adversarial contrastive risk, the adversary is restricted to perturbing only the anchor sample $x$, while the positive sample $x^+$ and negative samples $x^-$ remain clean. This is consistent with standard practices in both theoretical (Zou & Liu, 2023) and practical applications of ACL (Kim et al., 2020; Ho & Nvasconcelos, 2020; Jiang et al., 2020), where adversarial perturbations are typically applied to the anchor sample alone. This setup allows us to evaluate the robustness of the learned representation without disrupting the fundamental contrastive structure between positive and negative samples.

*Our goal is to derive a generalization bound for ACL, that is a bound on $L_{\text{un}}^{\text{adv}}(f) - \hat{L}_{\text{un}}^{\text{adv}}(f)$.*

## 4 Generalization Error Bounds

In this section, we establish a generalization bound for ACL. Our technique relies on the concept of *covering numbers* of the adversarial contrastive loss class. *Covering numbers* measure the complexity of a function class $\mathcal{F}$ by counting the minimum number of "balls" needed to cover all functions in $\mathcal{F}$, where each ball represents a region of approximation with a specified level of accuracy.

**Definition 4.1** (Covering number). Let $\mathcal{F} := \{(f(x_1), \cdots, f(x_n))\}$ be a real-valued function class, that maps $\mathcal{X} \to \mathbb{R}^d$ defined over a vector space $\mathcal{V}$, and let $S := \{x_1, \cdots, x_n\} \subset \mathcal{X}^n$ be a dataset. For any $\epsilon > 0$, the $\ell_p$-norm covering number, denoted as $\mathcal{N}_p(\epsilon, \mathcal{F}, S)$, is defined as the size of the smallest set of vectors $v_1, \cdots, v_m$ that covers $\mathcal{F}$. Specifically, it satisfies:

$$\sup_{f \in \mathcal{F}} \min_{j \in [m]} \left( \frac{1}{n} \sum_{i \in [n]} |f(x_i) - v_j^i|^p \right)^{\frac{1}{p}} \le \epsilon,$$

where $v_1, \cdots, v_m$ forms the $(\epsilon, \ell_p)$-cover of $\mathcal{F}$ with respect to $S$. Moreover, when the $\ell_p$-norm is taken as the $\ell_\infty$-norm, we denote the covering number as $\mathcal{N}_\infty(\epsilon, \mathcal{F}, S)$. Finally, the worst-case covering number is defined as $\mathcal{N}_p(\epsilon, \mathcal{F}, n) = \max_{S \in \mathcal{X}^n} \mathcal{N}_p(\epsilon, \mathcal{F}, S)$, where the maximum is taken over all possible datasets $S \subset \mathcal{X}^n$ of size $n$.

To analyze the complexity of a function class involving a nonlinear loss function (e.g., hinge loss), we use Lipschitz continuity to simplify the analysis by reducing the complexity to that of a function class without the loss function $\ell$. We consider a general Lipschitz continuity w.r.t. a $\ell_p$-norm, defined as follows:

**Definition 4.2** (Lipschitz continuity). A function $\ell : \mathbb{R}^C \to \mathbb{R}_+$ is said to be $\lambda$-Lipschitz continuous with respect to the $\ell_p$-norm ($p \ge 1$) if for any $v, v' \in \mathbb{R}^C$, the following inequality holds:

$$|\ell(v) - \ell(v')| \le \lambda \|v - v'\|_p.$$

In other words, a Lipschitz continuous function exhibits only a small change in its output when its inputs slightly change.

We now proceed to derive the generalization bounds for ACL. The function class of interest is defined as follows:

$$\mathcal{G}_{\text{adv}} = \left\{ (x, x^+, x_1^-, \ldots, x_K^-) \mapsto \max_{\delta \in \mathcal{B}} \ell(\{f(A(x, \delta)^T (f(x^+) - f(x_k^-))\}_{k=1}^{K}) : f \in \mathcal{F} \right\}.$$

We use covering numbers to measure the complexity of our function class $\mathcal{G}_{\text{adv}}$ by quantifying how well it can be approximated using simpler functions. This approximation helps establish generalization bounds, as a smaller covering number indicates a less complex function class, leading to tighter bounds on the adversarial contrastive loss.

The structure of the analysis is as follows: 1) The Lemma 4.1 simplifies the adversarial contrastive learning function class ($\mathcal{G}_{\text{adv}}$) by leveraging the Lipschitz continuity of the loss function ($\lambda_2$) and $\max_{\delta \in \mathcal{B}}$ to reduce its complexity to that of an intermediate class, $\mathcal{H}$. 2) Building on this, the Lemma 4.2 further reduces $\mathcal{H}$ to an even simpler function class, $\tilde{\mathcal{F}}$. 3) This final simplified function class, $\tilde{\mathcal{F}}$, forms the basis of our main Theorem 4.1, which depicts that the generalization error of ACL is governed by the covering number of this reduced function class.

The main challenge for the analysis of the ACL function class is due to the $\max_{\delta \in \mathcal{B}}$ operator. The $\max_{\delta \in \mathcal{B}}$ operator adds complexity by requiring optimization over all perturbations, making the function class hard to analyze. To address this, our main strategy is to leverage the properties of $\ell_\infty$-covering numbers to control the complexity of the ACL function class. This approach allows us to effectively handle the high-dimensional function class and alleviate the challenges posed by the maximization operator (Mustafa et al., 2022). Removing this operator simplifies the class, enabling tractable analysis and the derivation of generalization bounds.

The following lemma shows a bound on the covering number of $\mathcal{G}_{\text{adv}}$ in terms of the covering number of an extended function class that does not contain the $\max_{\delta \in \mathcal{B}}$-operator and the loss function $\ell$ on $f$. The extended function class is defined as:

$$\mathcal{H} = \left\{ (x, x^+, x^-, \tilde{\delta}) \mapsto f(A(x, \tilde{\delta}))^T (f(x^+) - f(x^-)) : f \in \mathcal{F} \right\}.$$

That is, the functions in $\mathcal{H}$ are explicitly parameterized by the adversarial noise $\delta$. Consequently, the data set is extended to:

$$S_{\mathcal{H}} = \left\{ (x_i, x_i^+, x_{ik}^-, \tilde{\delta}) : i \in [n], k \in [K], \tilde{\delta} \in \mathcal{C}_{\mathcal{B}}(\frac{\epsilon}{2\lambda_1}) \right\},$$

where $\mathcal{C}_{\mathcal{B}}(\frac{\epsilon}{2\lambda_1})$ is an $(\frac{\epsilon}{2\lambda_1}, \ell_\infty)$-cover of $\mathcal{B}$, for some $\epsilon, \lambda_1 > 0$. Specifically, for any $\delta \in \mathcal{B}$, there exists $\tilde{\delta} \in \mathcal{C}_{\mathcal{B}}(\frac{\epsilon}{2\lambda_1})$ such that $\|\delta - \tilde{\delta}\|_\infty \leq \frac{\epsilon}{2\lambda_1}$. We now introduce our first lemma, which establishes a relationship between the covering number of $\mathcal{G}_{\text{adv}}$ on $S$ and that of $\mathcal{H}$ on $S_{\mathcal{H}}$.

**Lemma 4.1.** Let $\delta \mapsto \ell(\{f(A(x, \delta))^T (f(x^+) - f(x_k^-))\}_{k=1}^K)$ be $\lambda_1$-Lipschitz and $\ell$ be $\lambda_2$-Lipschitz with respect to the $\ell_\infty$-norm, for all $(x, x^+, x_1^-, \ldots, x_K^-) \in \mathcal{X}^{K+2}$ and $f \in \mathcal{F}$. Then, we have:

$$\mathcal{N}_\infty(\epsilon, \mathcal{G}_{\text{adv}}, S) \leq \mathcal{N}_\infty\left( \frac{\epsilon}{2\lambda_2}, \mathcal{H}, S_{\mathcal{H}} \right).$$

The lemma shows that we can upper-bound the $\ell_\infty$-covering number of the ACL function class by that of the class $\mathcal{H}$ with the extended training set $S_{\mathcal{H}}$. Notably, the class $\mathcal{H}$ does not include the $\max_{\delta \in \mathcal{B}}$ operator, significantly simplifying the analysis. Furthermore, it shifts the dependence on the number of negative samples from the dimensionality of the function class's output to the size of the training set. For most classes, the dependence of the covering numbers on the size of the training set is only logarithmic (Zhang, 2002). Consequently, our bound will lead to a generalization bound that only has a logarithmic dependence on the number of negative samples $K$. The proof of Lemma 4.1 begins by eliminating the $\max_{\delta \in \mathcal{B}}$-operator, an approach based on Mustafa et al. (2022). Details of the proof are provided in the appendix.

*Remark.* The Lipschitz condition on the function $\delta \mapsto \ell(\{f(A(x, \delta))^T (f(x^+) - f(x_k^-))\}_{k=1}^K)$ is crucial because it is a mild yet standard assumption that most adversarial attacks in the literature satisfy (Engstrom et al., 2019; Awasthi et al., 2021; Madry et al., 2017). This condition ensures that the loss function behaves smoothly and predictably, which is essential for the success of gradient-based adversarial attacks. Additionally, this Lipschitzness allows us to bound the covering number of the adversarial class $\mathcal{G}_{\text{adv}}$.

*Remark.* The size of the extended training set $S_{\mathcal{H}}$ grows linearly with the size of the cover $\mathcal{C}_{\mathcal{B}}(\frac{\epsilon}{2\lambda_1})$. While the size of $\mathcal{C}_{\mathcal{B}}(\frac{\epsilon}{2\lambda_1})$ can grow exponentially with the dimensionality of the perturbation set $\mathcal{B}$, it's important to note that the dependence of the generalization performance is typically of the order $\mathcal{O}(\log^{1/2}(|S_{\mathcal{H}}|))$, as shown in prior work (Bartlett et al., 2017; Zhang, 2002; Mustafa et al., 2021). Thus, the generalization bounds will exhibit a square-root dependency on the dimensionality of $\mathcal{B}$, leading to manageable bounds even in the presence of large perturbation sets.

While Lemma 4.1 provides an upper bound on the ACL class $\mathcal{G}_{\text{adv}}$ in terms of the non-adversarial class $\mathcal{H}$, the class $\mathcal{H}$ is not directly the representation function class $\mathcal{F}$. This makes it challenging to utilize existing

covering number results for typical models (e.g., linear models (Zhang, 2002) or DNNs (Ledent et al., 2021b)). The following lemma establishes a relationship between the covering numbers of $\mathcal{H}$ and those of $\mathcal{F}$.

**Lemma 4.2.** Assume the previous conditions hold and that $\|f(x)\|_1 \leq R$, for all $x \in \mathcal{X}$. We define the function class $\tilde{\mathcal{F}}$ as follows:

$$\tilde{\mathcal{F}} = \big\{(x,j) \mapsto f_j(x) : f \in \mathcal{F}, x \in \mathcal{X}, j \in [d]\big\}$$

over the training set $S_{\tilde{\mathcal{F}}}$:

$$S_{\tilde{\mathcal{F}}} = \{(A(x_i, \tilde{\delta}), j) : i \in [n], j \in [d], \tilde{\delta} \in \mathcal{C}_{\mathcal{B}}(\frac{\epsilon}{2\lambda_1})\} \cup \{(x_{ik}^-, j) : i \in [n], k \in [K], j \in [d]\} \cup \{(x_i^+, j) : i \in [n], j \in [d]\}.$$

Then, we have:

$$\mathcal{N}_\infty\Big(\frac{\epsilon}{2\lambda_2}, \mathcal{H}, S_{\mathcal{H}}\Big) \leq \mathcal{N}_\infty\Big(\frac{\epsilon}{8R\lambda_2}, \tilde{\mathcal{F}}, S_{\tilde{\mathcal{F}}}\Big).$$

The proof of this lemma is provided in the appendix. The lemma upper-bounds the covering number of $\mathcal{H}$ by the covering number of the class $\tilde{\mathcal{F}}$. Notably, $\tilde{\mathcal{F}}$ is a class of scalar-valued functions of the same form as the representation function class $\mathcal{F}$. This simplifies the analysis by (1) reducing the form of the functions in $\mathcal{H}$ to that of the representation class, and (2) simplifying the analysis from vector-valued functions to scalar functions of the same form. Note that the number of dimensions contributes only through the size of the dataset $S_{\tilde{\mathcal{F}}}$, and for many typical function classes, this contribution is only logarithmic. This achieves the best known rate for vector-valued functions (Lei et al., 2019). Combining Lemmas 4.2 and 4.1 with Dudley's entropy integral (Boucheron et al., 2003; Bartlett et al., 2017; Ledent et al., 2021a; Srebro et al., 2010) gives our main result.

**Theorem 4.1.** Let $\delta \in (0,1)$, and $\mathcal{F} = \{f : \|f(\cdot)\|_1 \leq R\}$, for some $R > 0$, where $\|\cdot\|_1$ denotes the $\ell_1$-norm, and $d \in \mathbb{N}$ represents the dimensionality of the feature space and $\tilde{\mathcal{F}} = \big\{(x,j) \mapsto f_j(x) : f \in \mathcal{F}, x \in \mathcal{X}, j \in [d]\big\}$. With probability at least $1 - \delta$ over the randomness of the training data $S$ with size $n$, we have for all $f \in \mathcal{F}$:

$$L_{\text{un}}^{\text{adv}}(f) \leq \hat{L}_{\text{un}}^{\text{adv}}(f) + \frac{3B\sqrt{\log(\frac{2}{\delta})}}{\sqrt{2n}} + \inf_{a>0}\Big(8a + \frac{24}{\sqrt{n}}\int_a^B \log^{\frac{1}{2}}\mathcal{N}_\infty\big(\frac{\epsilon}{8R\lambda_2}, \tilde{\mathcal{F}}, S_{\tilde{\mathcal{F}}}\big)d\epsilon\Big).$$

The theorem demonstrates that we can control the generalization error of ACL by controlling the covering number of the class $\tilde{\mathcal{F}}$. The covering numbers of many classes $\tilde{\mathcal{F}}$ (e.g., linear models (Zhang, 2002), MLPs (Bartlett et al., 2017), CNNs (Ledent et al., 2021b), and structured learning models (Mustafa et al., 2021)) can be directly applied here to derive generalization bounds for ACL across a large family of models.

## 5 Applications

To learn the representations $f$ in an unsupervised setting, we employ an unsupervised loss function $\ell : \mathbb{R}^K \mapsto \mathbb{R}_+$, which can be selected as the hinge loss. For simplicity, we assume the loss function is bounded by $B$ for any $f \in \mathcal{F}$. Specifically, this means:

$$\ell(\{f(A(x,\delta))^T(f(x^+) - f(x_k^-))\}_{k=1}^K) \leq B, \quad \forall f \in \mathcal{F}.$$

This assumption is valid because we can impose constraints on the norms of the model's weights and inputs, ensuring the loss function remains bounded.

In this section, we instantiate our bound (Theorem 4.1) for two models: linear and DNN-based features. Throughout the section, we consider feature extractors of the form $x \mapsto U\mathbf{v}(x)$, where $U \in \mathbb{R}^{d \times d'}$ is a transformation matrix, and $\mathbf{v} : \mathcal{X} \mapsto \mathbb{R}^{d'}$ is a map from the original data $x \in \mathcal{X}$ to some intermediate embedding space in $\mathbb{R}^{d'}$. We consider the linear feature extractor in Section 5.1, while in Section 5.2, we explore features from a DNN.

### 5.1 Linear Features

First, we focus on the linear features. That is, we assume that $\mathbf{v}$ is the identity map.

We consider the $\ell_\infty$-attack, in which the attacker uses an additive noise function $A(x, \delta) = x + \delta$, for $x \in \mathcal{X}$ and $\delta \in \mathcal{B}$, where the noise set, $\mathcal{B}$, is the $\ell_\infty$-ball

$$\mathcal{B} = \left\{ \delta : \|\delta\|_\infty \leq \beta \right\} \subset \mathbb{R}^D.$$

We begin by showing that the function $\delta \mapsto \ell((U\mathbf{v}(x+\delta))^T (U\mathbf{v}(x^+) - U\mathbf{v}(x_k^-))_{k=1}^K)$ is indeed Lipschitz. The following lemma establishes and quantifies the upper bound on the Lipschitz constant of $\delta \mapsto \ell((U(x+\delta))^T (U(x^+) - U(x_k^-))_{k=1}^K)$.

**Lemma 5.1.** Consider the function $g_U(x, \delta) = \ell((U(x+\delta))^T (U(x^+) - U(x_k^-))_{k=1}^K)$ and assume $\|U\|_{\infty,2} \leq \Lambda_1$. Then, for any $x$, the function $\delta \mapsto g_U(x, \delta)$ is $\|\cdot\|_\infty$-Lipschitz with the Lipschitzness constant $2\Lambda_1^2 \|x\|_2$.

Now that we have the Lipschitzness of $\delta$ on the loss function $\ell$, we can bound the covering number of our linear scalar-valued feature class in the following lemma.

**Lemma 5.2.** Let $\tilde{\mathcal{F}}$ be the linear feature class and $S_{\tilde{\mathcal{F}}}$ be a given dataset in Lemma 4.2 with $\|x\|_2 \leq \Psi$, for all $x \in \mathcal{X}$, and $\|U\|_{2,2} \leq \Lambda$, then for all $\epsilon > 0$, we have

$$\log \mathcal{N}_\infty \left( \frac{\epsilon}{8R\lambda_2}, \tilde{\mathcal{F}}, S_{\tilde{\mathcal{F}}} \right) \leq \frac{CR^2 \lambda_2^2 \Lambda^2 (\Psi + \sqrt{D}\beta)^2 L_{\log}}{\epsilon^2},$$

where $C$ is an absolute constant, $m = |\mathcal{C}_\mathcal{B}(\frac{\epsilon}{2\lambda_1})|$, $\Psi' = \Psi + \sqrt{D}\beta$ and

$$L_{\log} := \log \left( 2 \left\lceil \frac{8R\lambda_2 \Lambda \Psi'}{\epsilon} + 2 \right\rceil (nmd + ndK + nd) \left( \frac{12\beta \Lambda_1^2 \|x\|_2 \Psi}{\epsilon} \right)^D + 1 \right).$$

If we plug Lemma 5.2 back into the Theorem 4.1, we get the following corollary.

**Corollary 5.1.** Assuming the above assumptions, for all $f \in \mathcal{F}$, with probability at least $1 - \delta$ over the training data, we have

$$L_{\mathrm{un}}^{\mathrm{adv}}(f) \leq \hat{L}_{\mathrm{un}}^{\mathrm{adv}}(f) + \frac{8}{n} + 3B\sqrt{\frac{\log(2/\delta)}{2n}} + \frac{\sqrt{C}R\lambda_2 \Lambda \Psi' \tilde{L}_{\log}}{\sqrt{n}},$$

where $C$ is a constant, $\Psi' = \Psi + \sqrt{D}\beta$, $m = |\mathcal{C}_\mathcal{B}(\frac{\epsilon}{2\lambda_1})|$, $N = nmd + ndK + nd$ and

$$\tilde{L}_{\log} := \log^{\frac{1}{2}} \left( 4 \lceil 4R\lambda_2 \Lambda \Psi' n + 1 \rceil N \left( 12\beta \Lambda_1^2 \|x\|_2 \Psi n \right)^D + 1 \right) (\log(n) + \log(B)).$$

*Remark.* Our bound has a dependency on the square root of the input dimension, $\sqrt{D}$, in the term $\Psi'$. This arises due to the mismatch between the $\ell_2$-norm of the input and the $\ell_\infty$-norm in the ball $\beta$, as encapsulated by the inequality $\|\delta\|_2 \leq \sqrt{D}\|\delta\|_\infty$. Additionally, the bound has a logarithmic dependence on the negative samples, $K$. The logarithmic dependency shows the appealing behavior of ACL for learning with a large number of negative examples.

## 5.2 Nonlinear Features

Now, we consider the covering numbers for learning the nonlinear features by DNNs. We say an activation function $\sigma : \mathbb{R} \mapsto \mathbb{R}$ is positive-homogeneous if $\sigma(ax) = a\sigma(x)$ for $a > 0$, and is contracting if $|\sigma(x) - \sigma(x')| \leq |x - x'|$. The ReLU activation function $\sigma(x) = \max\{x, 0\}$ is both positive-homogeneous and contractive. Now assume the DNN feature map is defined as (removing matrix $U$ for now),

$$\mathcal{V} = \{x \mapsto \mathbf{v}(x) = \sigma(V_L \sigma(V_{L-1} \cdots \sigma(V_1 x))) : \forall l \in [L]\}.$$

Each layer $l \in [L]$ has the width of $w_l$, where $w_0 = D$ (the input dimension) and $w_L = d$ (the number of feature dimension).

Let $V \in \mathbb{V}$ be the weight of the network. Suppose that $\mathbb{V}$ is such that, for all $V \in \mathbb{V}$, $\|V_l\|_2 \leq a_l$ and $\|V_l\|_\sigma \leq s_l$ for all $l \in [L-1]$. Further, suppose that, for all $V \in \mathbb{V}$, $\|V_L\|_2 \leq a_L$, $\|V_L\|_{2,\infty} \leq s_L$ and $\|V_1\|_{1,\infty} \leq s_1'$.

We now consider the $\ell_\infty$-additive perturbation applied to the DNN. As with the linear case, we first establish the Lipschitzness of the function $\delta \mapsto \ell((U\mathbf{v}(x+\delta))^T(U\mathbf{v}(x^+) - U\mathbf{v}(x_k^-))_{k=1}^K)$ w.r.t. $\|\cdot\|_\infty$-norm. The following lemma establishes the Lipschitz continuity of the loss as a function of $\delta$.

**Lemma 5.3.** Consider the function $g_{UV}(x,\delta) = \ell((U\mathbf{v}(x+\delta))^T(U\mathbf{v}(x^+) - U\mathbf{v}(x^-))_{k=1}^K)$ and assume $\|U\|_{\infty,2} \leq \Lambda_1$ and $\mathbf{v}(\cdot)$ is the neural network. Then, for any $x \in \mathcal{X}$ and $V \in \mathbb{V}$, the function $\delta \mapsto g_{UV}\mathbf{v}(x,\delta)$ is $\|\cdot\|_\infty$-Lipschitz with constant $2\Lambda_1^2 s_1'\sqrt{w_1}\|x\|_2 \prod_{l=1}^L s_l \prod_{l=2}^L s_l$.

Now, we can get the upper bound of the covering number of the neural network scalar-valued feature class w.r.t. $\|.\|_\infty$-norm.

**Lemma 5.4.** Let $\tilde{\mathcal{F}}$ be the DNN (nonlinear) feature class on the extended dataset $S_{\tilde{\mathcal{F}}}$, defined as before. Let $\mathcal{B} := \{\delta : \|\delta\|_\infty \leq \beta\}$. Assume the previous assumptions on the weights of DNN, and $\|x\|_2 \leq \Psi$. Then, for $S_{\tilde{\mathcal{F}}}$ and $\epsilon > 0$, we have

$$\log \mathcal{N}_\infty\left(\frac{\epsilon}{8R\lambda_2}, \tilde{\mathcal{F}}, S_{\tilde{\mathcal{F}}}\right) \leq \frac{CL^2R^2\lambda_2^2\Psi'^2}{\epsilon^2}\prod_{l=1}^L s_l^2 \left(\sum_{l=1}^L \frac{a_l^2}{s_l^2}\right) L_{\log},$$

where

$$L_{\log} := \log\left(\left(C_1\Psi'\Gamma R^2\lambda_2/\epsilon + C_2\bar{w}\right)(nmd + ndK + nd)\left(\frac{12\beta\Lambda_1^2 s_1'\sqrt{w_1}\|x\|_2 \prod_{l=1}^L s_l \prod_{l=2}^L s_l}{\epsilon}\right)^D + 1\right),$$

$\Psi' = (\Psi + \sqrt{D}\beta)$, $\Gamma = \max_{l \in [L]}(\prod_{i=1}^L s_i)a_l w_l/s_l$, $\bar{w} = \max_{l \in [L]} w_l$, $m = |\mathcal{C}_\mathcal{B}(\frac{\epsilon}{2\lambda_1})|$, and $C, C_1, C_2$ are universal constants.

Plugging Lemma 5.4 into Theorem 4.1, we get the following corollary.

**Corollary 5.2.** Under the previous assumptions, for all $f \in \mathcal{F}$, with probability at least $1 - \delta$ over the training data, we have

$$L_{\mathrm{un}}^{\mathrm{adv}}(f) \leq \hat{L}_{\mathrm{un}}^{\mathrm{adv}}(f) + \frac{8}{n} + 3\sqrt{\frac{\log(2/\delta)}{2n}} + \frac{CLR\lambda_2\Psi'}{\sqrt{n}}\prod_{l=1}^L s_l \sqrt{\left(\sum_{l=1}^L \frac{a_l^2}{s_l^2}\right)}\tilde{L}_{\log},$$

where $C$ is an absolute constant, $\Psi' = \Psi + \sqrt{D}\beta$, $m = |\mathcal{C}_\mathcal{B}(\frac{\epsilon}{2\lambda_1})|$, $N = nmd + ndK + nd$ and

$$\tilde{L}_{\log} = \log^{\frac{1}{2}}\left(\left(C_1\Psi'\Gamma n + C_2\bar{w}\right)N\left(12\beta\Lambda_1^2 s_1'\sqrt{w_1}\|x\|_2\prod_{l=1}^L s_l \prod_{l=2}^L s_l n\right)^D + 1\right)(\log(n) + \log(B)).$$

*Remark.* Similar to the results for the linear case, our analysis reveals a dependency on the square root of the input dimension. This issue can be resolved if we assume the $\ell_2$ attack, where $\mathcal{B} = \{\delta : \|\delta\|_2 \leq \beta\}$. As in the linear case, our results maintain a logarithmic dependence on the negative samples, $K$.

# 6 Experiments

We evaluate our theoretical results to two widely used benchmark datasets from the image domain: CIFAR-10 and CIFAR-100 (Krizhevsky, 2009). CIFAR-10 and CIFAR-100 consist of 50,000 training images and 10,000 testing images, organized into 10 and 100 classes, respectively.

**Experimental Setup** For linear features, we use a one-layer neural network, while for nonlinear features, we implement a four-layer neural network with ReLU activation. Both models are trained using the Adam optimizer, with a learning rate of $1e - 3$. We perform adversarial training on the following objective function:

$$\arg\max_{\delta \in \mathcal{B}} \ell(\{f(A(x,\delta))^T(f(x^+) - f(x_k^-))\}_{k=1}^k).$$

To generate adversarial perturbations, we employ the $\ell_\infty$ PGD algorithm (Madry, 2017) with a step size of $\epsilon/255$, where $\epsilon$ represents the maximum allowable perturbation. Afterward, a PGD attack is performed to evaluate the generalization error. The generalization error is calculated as the difference between the training accuracy and testing accuracy, a standard method commonly used in the literature Yin et al. (2019). We calculate the generalization error for different values of $\epsilon = \{2/255, 4/255, 8/255, 16/255, 32/255, 64/255, 128/255\}$, with varying negative samples, $K = \{63, 127, 511, 1023\}$.

## 6.1 Results

We analyze how the generalization error varies with respect to different parameters by examining the effect of step size ($\epsilon$) and the number of negative samples ($K$). The results are presented in Figure 1 for CIFAR-10 and CIFAR-100. As expected, the generalization error increases as the number of negative samples ($K$) grows. While this behavior is consistent with our theoretical bounds, it contrasts with findings in prior work showing that increasing $K$ can enhance downstream classification performance (Wang et al., 2022; Bao et al., 2022; Awasthi et al., 2022). This apparent discrepancy arises from the different objectives in these two types of analyses. The generalization error increases with $K$ because the function class becomes more complex, as the loss function must handle a greater number of comparisons. Furthermore, a larger $K$ can make the model more prone to overfitting to the negative examples, thereby amplifying the generalization gap—particularly if the loss function becomes overly sensitive to the contrast between positive and negative pairs. These two phenomena are not inherently contradictory but rather highlight the distinction between task-specific performance and the broader ability of the learned representations to generalize across diverse tasks.

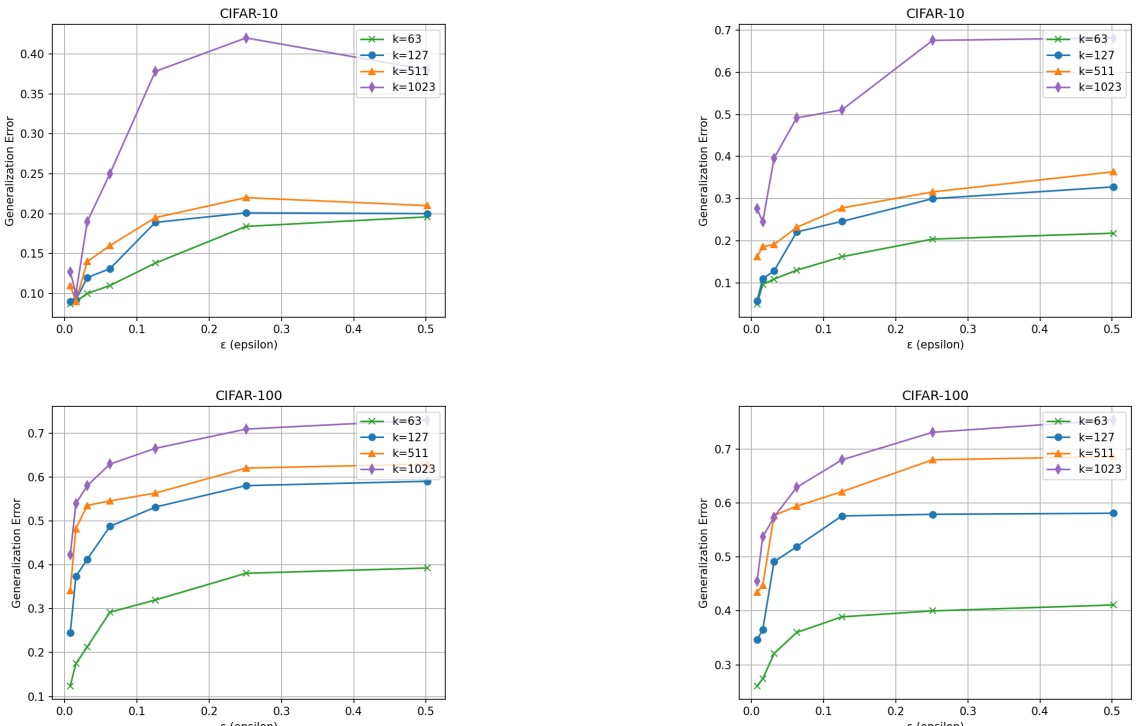

Figure 1: Generalization error with varying numbers of negative samples. On the left, the features are learned using a linear model, and on the right, the features are learned using a nonlinear model (4-layer neural network). As the number of negative samples ($K$) increases, the generalization error rises.

## 7 Proofs

### 7.1 Proofs of Results in Section 5.1

In this subsection, we present the omitted proofs of section 5.1 when the features are linear. Our approach relies on $\ell_\infty$-covering numbers for the feature classes. First, we show that the loss function is $\ell_\infty$-Lipschtiz with respect to the noise parameter $\delta$. Next, we derive a bound on the size of the set $\mathcal{C}_\mathcal{B}(\epsilon/2\lambda_1)$. Finally, we establish a bound on the $\ell_\infty$-covering number of the feature class $\tilde{\mathcal{F}}$ on the extended data set $S_{\tilde{\mathcal{F}}}$.

Initially, we prove the bounds of $\ell_\infty$-additive attacks applied to linear models as stated in Lemma 5.1. Our first step is to derive the $\|\cdot\|_\infty$-Lipschitz constant of the function $\delta \mapsto \ell(\{(U(x+\delta))^T(U(x^+) - U(x_k^-))\}_{k=1}^K)$. The following is the proof of Lemma 5.1.

*Proof of Lemma 5.1.* The proof is a direct derivation. For all $x \in \mathcal{X}$, $\|U\|_{\infty,2} \le \Lambda_1$, $\delta, \delta' \in \mathcal{B}$, we have:

$$
\begin{aligned}
&|\ell(\{(U(x+\delta))^T(U(x^+) - U(x_k^-))\}_{k=1}^K) - \ell(\{(U(x+\delta'))^T(U(x^+) - U(x_k^-))\}_{k=1}^K)| \\
&\le |\ell((U(x+\delta))^T(U(x^+) - U(x^-))) - \ell((U(x+\delta'))^T(U(x^+) - U(x^-)))| \\
&\le |(U(x+\delta))^T(U(x^+) - U(x^-)) - (U(x+\delta'))^T(U(x^+) - U(x^-))| \\
&\le \|U(x+\delta) - U(x+\delta')\|_2 \|U(x^+) - U(x^-)\|_2 \le \Lambda_1 \|(x+\delta) - (x+\delta')\|_\infty \|U(x^+) - U(x^-)\|_2 \\
&\le \Lambda_1 \|\delta - \delta'\|(\|U(x^+)\|_2 + \|U(x^-)\|_2) \le \Lambda_1 \|\delta - \delta'\|_\infty 2\Lambda_1 \|x\|_\infty \le 2\Lambda_1^2 \|x\|_2 \|\delta - \delta'\|_\infty.
\end{aligned}
$$

The second inequality is derived from the fact that hinge loss $\ell$ is $\ell_\infty$-Lipschitz with constant 1. The fifth inequality uses the triangle inequality $\|U(x^+) - U(x^-)\|_2 \le \|U(x^+)\|_2 + \|U(x^-)\|_2$. The last inequality follows from $\|x\|_\infty \le \|x\|_2$, for all $x \in \mathbb{R}^D$. $\square$

In this paper, we need upper bounds on covering numbers of bounded balls in $\mathbb{R}^D$. We start by reviewing a result that provides an upper bound on the size of the set $\mathcal{C}_\mathcal{B}(\epsilon)$ defined w.r.t. a general norm $\|\cdot\|$.

**Lemma 7.1** (Long & Sedghi 2019)**.** Let $d$ be a positive integer, $\|\cdot\|$ be a norm, $\rho$ be the metric induced by it, and $\kappa, \epsilon > 0$. A ball of radius $\kappa$ in $\mathbb{R}^d$ w.r.t. $\rho$ can be covered by $(\frac{3\kappa}{\epsilon})^d$ balls of radius $\epsilon$.

We now review the upper bounds on the $\ell_\infty$-covering numbers of linear models.

**Lemma 7.2** (Zhang 2002)**.** Let $\mathcal{L}$ be a class of linear functions on a set of size $n$. That is, $\mathcal{L} = \{\langle w, x \rangle, x, w \in \mathbb{R}^N\}$. If $\|x\|_q \le b$ and $\|w\|_p \le a$, where $2 \le q < \infty$ and $1/p + 1/q = 1$, then for any $\epsilon > 0$, we have

$$
\log \mathcal{N}_\infty(\epsilon, \mathcal{L}, n) \le 36(q-1)\frac{a^2 b^2}{\epsilon^2} \log[2\lceil 4ab/\epsilon + 2\rceil n + 1],
$$

where $\mathcal{N}_\infty(\epsilon, \mathcal{L}, n)$ is the worst-case covering number of the class $\mathcal{L}$ on a dataset of size $n$.

In the following, we present the proof of Lemma 5.2.

*Proof of Lemma 5.2.* First, we consider the $\ell_\infty$-norm on the set $\mathcal{B}$. According to Lemma 5.1, the function $\delta \mapsto \ell(\{(U(x+\delta))^T(U(x^+) - U(x_k^-))\}_{k=1}^K)$ is $\|\cdot\|_\infty$-Lipschitz with a constant of $2\Lambda_1^2 \|x\|_2$. Next, consider the set $\mathcal{C}_\mathcal{B}(\epsilon/4\Lambda_1^2 \|x\|_2)$. By applying Lemma 7.1, and noting that $\|\delta\|_\infty \le \beta$, we have for all $\delta \in \mathcal{B}$:

$$
\left|\mathcal{C}_\mathcal{B}(\epsilon/4\Lambda_1^2 \|x\|_2)\right| \le \left(\frac{12\Lambda_1^2 \|x\|_2 \beta}{\epsilon}\right)^D.
$$

Thus, the size of our dataset is:

$$
|S_{\tilde{\mathcal{F}}}| = n\left(\frac{12\Lambda_1^2 \|x\|_2 \beta}{\epsilon}\right)^D d + ndK + nd.
$$

For $\tilde{x} \in S_{\tilde{\mathcal{F}}}$, where $\tilde{x} = (x, \tilde{\delta})$, we have: $\|\tilde{x}\|_2 \le \|x\|_2 + \|\tilde{\delta}\|_2 \le \Psi + \sqrt{D}\|\delta\|_\infty = \Psi'$. Therefore, the result follows from Lemma 7.2. $\square$

Below, we provide the proof for Corollary 5.1.

*Proof of Corollary 5.1.* The proof follows directly from Theorem 4.1 by setting $\alpha$ to $\frac{1}{n}$. Therefore, consider the following integral

$$\int_a^B \sqrt{\log \mathcal{N}_\infty\left(\frac{\epsilon}{8R\lambda_2}, \tilde{\mathcal{F}}, S_{\tilde{\mathcal{F}}}\right)} d\epsilon \leq \int_{\frac{1}{n}}^B \sqrt{\frac{CR^2\lambda_2^2\Lambda^2(\Psi + \sqrt{D}\beta)^2 L_{\log}}{\epsilon^2}} d\epsilon \leq \sqrt{C}R\lambda_2\Lambda\Psi'\frac{\tilde{L}_{\log}}{\log(n)}\int_{\frac{1}{n}}^B \frac{1}{\epsilon} d\epsilon$$

$$\leq \sqrt{C}R\lambda_2\Lambda\Psi'\frac{\tilde{L}_{\log}}{\log(n)}\left[\log(\epsilon)\right]_{\frac{1}{n}}^B \leq \sqrt{C}R\lambda_2\Lambda\Psi'\frac{\tilde{L}_{\log}}{\log(n)}(\log(B) + \log(n)).$$

The first inequality follows from the monotonicity property of integrals. The second inequality derives from the observation that replacing $\epsilon$ by $\frac{1}{n}$ in $\tilde{L}_{\log}$ can only increase its value. Substituting this into Theorem 4.1 yields the desired result. $\square$

## 7.2 Proofs of Results in Section 5.2

In this subsection, we provide the omitted proofs from section 5.2 for the case when the features are non-linear. As in the linear case, we begin by showing that the loss function is $\ell_\infty$-Lipschitz with respect to the noise parameter $\delta$. We then establish a bound on the set $\mathcal{C}_\mathcal{B}(\epsilon/2\lambda_1)$ and apply the $\ell_\infty$-covering number results of the non-linear feature class $\tilde{\mathcal{F}}$ on the extended dataset $S_{\tilde{\mathcal{F}}}$.

First, we prove the bounds of the $\ell_\infty$-additive attacks applied to non-linear models as stated in Lemma 5.3. The first step is to derive the $\ell_\infty$-Lipschitz constant of the function $\delta \mapsto \ell(\{(U\mathbf{v}(x+\delta))^T(U\mathbf{v}(x^+) - U\mathbf{v}(x_k^-))\}_{k=1}^K)$. The following is the proof of Lemma 5.3.

*Proof of Lemma 5.3.* The proof is a direct derivation. For all $x, x^+, x_k^- \in \mathcal{X}$, $\|U\|_{\infty,2} \leq \Lambda_1$, $\delta, \delta' \in \mathcal{B}$, we have:

$$|\ell(\{(U\mathbf{v}(x+\delta))^T(U\mathbf{v}(x^+) - U\mathbf{v}(x_k^-))\}_{k=1}^K) - \ell(\{(U\mathbf{v}(x+\delta'))^T(U\mathbf{v}(x^+) - U\mathbf{v}(x_k^-))\}_{k=1}^K)|$$

$$\leq |\ell((U\mathbf{v}(x+\delta))^T(U\mathbf{v}(x^+) - U\mathbf{v}(x^-))) - \ell((U\mathbf{v}(x+\delta'))^T(U\mathbf{v}(x^+) - U\mathbf{v}(x^-)))|$$

$$\leq |(U\mathbf{v}(x+\delta))^T(U\mathbf{v}(x^+) - U\mathbf{v}(x^-)) - (U\mathbf{v}(x+\delta'))^T(U\mathbf{v}(x^+) - U\mathbf{v}(x^-))|$$

$$\leq |(U\mathbf{v}(x+\delta) - U\mathbf{v}(x+\delta'))^T(U\mathbf{v}(x^+) - U\mathbf{v}(x^-))|$$

$$\leq \|U(\mathbf{v}(x+\delta) - \mathbf{v}(x+\delta'))\|_2 \|U(\mathbf{v}(x^+) - \mathbf{v}(x^-))\|_2$$

$$\leq \Lambda_1 \|\mathbf{v}(x+\delta) - \mathbf{v}(x+\delta')\|_\infty \|U(\mathbf{v}(x^+) - \mathbf{v}(x^-))\|_2,$$

where we have used the 1-Lipschitz property of the loss function $\ell$. Since $\|U\|_{\infty,2} \leq \Lambda_1$, we further get

$$|\ell(\{(U\mathbf{v}(x+\delta))^T(U\mathbf{v}(x^+) - U\mathbf{v}(x_k^-))\}_{k=1}^K) - \ell(\{(U\mathbf{v}(x+\delta'))^T(U\mathbf{v}(x^+) - U\mathbf{v}(x_k^-))\}_{k=1}^K)|$$

$$\leq \Lambda_1 \prod_{l=2}^L s_l \|V^1(x+\delta-x-\delta')\|_\infty \|U(\mathbf{v}(x^+) - \mathbf{v}(x^-))\|_2 \leq \Lambda_1 \prod_{l=2}^L s_l s_1' \|\delta - \delta'\|_2 \|U(\mathbf{v}(x^+) - \mathbf{v}(x^-))\|_2$$

$$\leq \Lambda_1 s_1' \prod_{l=2}^L s_l \sqrt{w_1} \|\delta - \delta'\|_\infty \|U(\mathbf{v}(x^+) - \mathbf{v}(x^-))\|_2 \leq \Lambda_1 s_1' \prod_{l=2}^L s_l \sqrt{w_1} \|\delta - \delta'\|_\infty \Lambda_1 \|\mathbf{v}(x^+) - \mathbf{v}(x^-)\|_\infty$$

$$\leq \Lambda_1 s_1' \prod_{l=2}^L s_l \sqrt{w_1} \|\delta - \delta'\|_\infty \Lambda_1 \prod_{l=1}^L s_l \|x^+ - x^-\|_\infty \leq \Lambda_1 s_1' \prod_{l=2}^L s_l \sqrt{w_1} \|\delta - \delta'\|_\infty \Lambda_1 2 \prod_{l=1}^L s_l \|x\|_2$$

$$\leq 2\Lambda_1^2 s_1' \sqrt{w_1} \|x\|_2 \prod_{l=1}^L s_l \prod_{l=2}^L s_l \|\delta - \delta'\|_\infty,$$

where we have used the 1-Lipschitz property of the non-linearity and induction over the layers. The eight inequality results from converting the $\|.\|_2$-norm to $\|.\|_\infty$-norm. Finally, the eleventh inequality is based on the fact that $\|x^+ - x^-\|_\infty \leq 2\|x\|_2$, for all $x \in \mathbb{R}^D$. $\square$

We now review the upper bounds on the $\ell_\infty$-covering numbers of norm-bounded neural networks (non-linear) function classes.

**Lemma 7.3** (Ledent et al. 2021b). Let $\mathcal{V}$ be the class of neural networks, that is, $\mathcal{V} = \{x \mapsto \mathbf{v}(x)\}$, where $V = (V^1, \ldots, V^L)$ are a set of weights the DNN $v(\cdot)$ and $\sigma$ is defined as above. Suppose that $\|V^l\|_{2,1} \le a_l$ and $\|V^l\|_\sigma \le s_l$ for all $l \in [L-1]$, $\|V^L\|_2 \le a_L$, $\|V^L\|_{2,\infty} \le s_L$, $\|x\|_2 \le b$, and $w_l$ is the width of the $l$'th layer. Then given a data set $S$ with $n$ elements and $\epsilon > 0$, we have

$$\log \mathcal{N}_\infty(\epsilon, \mathcal{V}, S) \le \frac{CL^2 b^2}{\epsilon^2} \prod_{l=1}^{L} s_l^2 \left( \sum_{l=1}^{L} \frac{a_l^2}{s_l^2} \right) \log\left(\left( C_1 b \Gamma / \epsilon + C_2 \bar{w} \right) n + 1 \right),$$

where $\Gamma = \max_{l \in [L]}(\prod_{i=1}^{L} s_i) a_l m_l / s_l$, $\bar{w} = \max_{l \in [L]} w_l$, and $C, C_1, C_2$ are universal constants.

In the following, we present the proof of Lemma 5.4.

*Proof of Lemma 5.4.* First, consider the $\ell_\infty$-norm on the set $\mathcal{B}$. By Lemma 5.3, we have the function $\delta \mapsto \ell(\{(U\mathbf{v}(x+\delta))^T (U\mathbf{v}(x^+) - U\mathbf{v}(x_k^-))\}_{k=1}^K)$ is $\|\cdot\|_\infty$-Lipschitz with constant $2\Lambda_1^2 s_1' \sqrt{w_1} \|x\|_2 \prod_{l=1}^{L} s_l \prod_{l=2}^{L} s_l$. Consider the set $\mathcal{C}_\mathcal{B}(\epsilon / 4\Lambda_1^2 s_1' \sqrt{w_1} \|x\|_2 \prod_{l=1}^{L} s_l \prod_{l=2}^{L} s_l)$. By Lemma 7.1, and that $\|\delta\|_\infty \le \beta$, we have for all $\delta \in \mathcal{B}$:

$$\left| \mathcal{C}_\mathcal{B}(\epsilon / 4\Lambda_1^2 s_1' \sqrt{w_1} \|x\|_2 \prod_{l=1}^{L} s_l \prod_{l=2}^{L} s_l) \right| \le \left( \frac{12 \Lambda_1^2 s_1' \sqrt{w_1} \|x\|_2 \prod_{l=1}^{L} s_l \prod_{l=2}^{L} s_l \beta}{\epsilon} \right)^D.$$

Thus, the size of our dataset is

$$|S_{\tilde{\mathcal{F}}}| = n \left( \frac{12 \Lambda_1^2 s_1' \sqrt{w_1} \|x\|_2 \prod_{l=1}^{L} s_l \prod_{l=2}^{L} s_l \beta}{\epsilon} \right)^D d + ndK + nd.$$

For $\tilde{x} \in S_{\tilde{\mathcal{F}}}$, where $\tilde{x} = (x, \tilde{\delta})$, we have

$$\|\tilde{x}\|_2 \le \|x\|_2 + \|\tilde{\delta}\|_2 \le \Psi + \sqrt{D} \|\delta\|_\infty = \Psi'.$$

Therefore, the result follows from Lemma 7.3. $\square$

In the following, we present the proof of Corollary 5.2.

*Proof of Corollary 5.2.* The proof is similar to the proof of Corollary 5.1. It is a direct application of Theorem 4.1 by setting $\alpha$ to $\frac{1}{n}$. $\square$

# 8 Conclusion

We conducted a generalization analysis of ACL, showing that the generalization error is bounded by the covering number of the feature class. Our results leverage the Lipschitz continuity and boundedness of the hinge loss as our unsupervised loss function, given the constraints on the model's weights and inputs. We applied this bound on both linear and non-linear features, subject to $\ell_\infty$-additive attacks. Unlike previous work, such as Zou & Liu (2023), our bounds are directly applied to the adversarial contrastive loss, avoiding the use of surrogate losses. Moreover, our bounds scale logarithmically with the number of negative samples $K$, with a complexity of $\mathcal{O}(\log K)$. Although these are algorithm-independent bounds, they could be extended to algorithm-dependent bounds to understand how the optimization process affects the generalization error. In this paper, we have applied only $\ell_\infty$-additive attacks; nonetheless, other types of adversarial attacks can also be tested, especially non-additive attacks.

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

# A  Appendix

In this section, we provide the missing proofs of section 4.

## A.1  Proofs of Lemma 4.1 and Lemma 4.2

To discretize the function space, we assume $\delta \mapsto \ell(\{f(A(x,\cdot))^T(f(x^+) - f(x_k^-))\}_{k=1}^K)$ is $\lambda_1$-Lipschitz for all $x \in \mathcal{X}$ and $f \in \mathcal{F}$. Let $\mathcal{C}_{\mathcal{B}}(\frac{\epsilon}{2\lambda_1})$ be an $(\frac{\epsilon}{2\lambda_1}, \ell_\infty)$-cover of $\mathcal{B}$. Assume the discretized class of loss function is $\tilde{\mathcal{G}}_{adv}$, defined as below:

$$\tilde{\mathcal{G}}_{\text{adv}} = \left\{(x, x^+, x_1^-, \ldots, x_K^-, \delta) \to \ell(\{f(A(x,\delta))^T(f(x^+) - f(x_k^-))\}_{k=1}^K) : f \in \mathcal{F}\right\}$$

with an extended training set $\tilde{S}$:

$$\tilde{S} = \left\{(x_i, x_i^+, x_{i1}^-, \ldots, x_{iK}^-, \tilde{\delta}) : i \in [n], \tilde{\delta} \in \mathcal{C}_{\mathcal{B}}\big(\epsilon/(2\lambda_1)\big)\right\}.$$

Now, we review a lemma from Mustafa et al. (2022) to relate the $\ell_\infty$-covering number of class $\mathcal{G}_{\text{adv}}$ to the covering number of the discretized version $\tilde{\mathcal{G}}_{adv}$.

**Lemma A.1** (Mustafa et al. 2022). Let $\tilde{\mathcal{G}}_{\mathrm{adv}}$ be defined as above. Then, the following holds:

$$\mathcal{N}_\infty(\epsilon, \mathcal{G}_{\mathrm{adv}}, S) \leq \mathcal{N}_\infty(\epsilon/2, \tilde{\mathcal{G}}_{\mathrm{adv}}, \tilde{S}).$$

This lemma simplifies the complexity of our function class by discretizing the loss function according to $\delta$.

Now, we can prove Lemma 4.1:

*Proof of Lemma 4.1.* According to Lemma A.1, it suffices to show that

$$\mathcal{N}_\infty(\epsilon/2, \tilde{\mathcal{G}}_{\mathrm{adv}}, \tilde{S}) \leq \mathcal{N}_\infty(\epsilon/(2\lambda_2), \mathcal{H}, S_{\mathcal{H}}).$$

The observation here is that we can construct a cover for the function class $\tilde{\mathcal{G}}_{\mathrm{adv}}$ on the training set $\tilde{S}$ from the elements of the cover of the function class $\mathcal{H}$. Additionally, from the Definition 4.1, we know that the covering number of a set is the cardinality of the smallest cover for a set.
For any $f$, define $h_f$ as

$$h_f(x, x^+, x^-, \tilde{\delta}) = f(A(x, \tilde{\delta}))^T (f(x^+) - f(x^-)).$$

Let $\mathcal{C}_B(\epsilon/(2\lambda_1)) = \{\delta_1, \ldots, \delta_m\}$. The projection of $\mathcal{H}$ onto the set $S_{\mathcal{H}}$ is

$$\mathcal{H}_{S_{\mathcal{H}}} := \left\{ \begin{bmatrix} h_f(x_1, x_1^+, x_{11}^-, \delta_1) & \cdots & h_f(x_1, x_1^+, x_{11}^-, \delta_m) \\ \vdots & \ddots & \vdots \\ h_f(x_n, x_n^+, x_{nK}^-, \delta_1) & \cdots & h_f(x_n, x_n^+, x_{nK}^-, \delta_m) \end{bmatrix} : f \in \mathcal{F} \right\} \subset \mathbb{R}^{nK \times m}.$$

Let

$$\mathcal{C}_{\mathcal{H}} := \left\{ \begin{bmatrix} c_{i'}^{11}(\delta_1) & \cdots & c_{i'}^{11}(\delta_m) \\ \vdots & \ddots & \vdots \\ c_{i'}^{nK}(\delta_1) & \cdots & c_{i'}^{nK}(\delta_m) \end{bmatrix} : i' = 1, \ldots, M \right\} \subset \mathbb{R}^{nK \times m}$$

be an $(\epsilon/(2\lambda_2), \ell_\infty)$-cover of $\mathcal{H}_{S_{\mathcal{H}}}$. This means, for all $f \in \mathcal{F}$, there exists an $r \in [M]$ such that

$$\max_{i \in [n]} \max_{k \in [K]} \max_{\delta \in \mathcal{C}_B(\frac{\epsilon}{2\lambda_1})} \left| h_f(x_i, x_i^+, x_{ik}^-, \delta) - c_r^{ik}(\delta) \right| \leq \frac{\epsilon}{2\lambda_2}.$$

Now we show the following set is an $(\epsilon/2, \ell_\infty)$-cover of $\tilde{\mathcal{G}}_{adv}$ w.r.t. $\tilde{S}$:

$$\mathcal{C}_{\tilde{\mathcal{G}}_{adv}} := \left\{ \begin{bmatrix} \ell(\{c_{i'}^{1k}(\delta_1)\}_{k=1}^K) & \cdots & \ell(\{c_{i'}^{1k}(\delta_m)\}_{k=1}^K) \\ \vdots & \ddots & \vdots \\ \ell(\{c_{i'}^{nk}(\delta_1)\}_{k=1}^K) & \cdots & \ell(\{c_{i'}^{nk}(\delta_m)\}_{k=1}^K) \end{bmatrix} : i' = 1, \ldots, M \right\} \subset \mathbb{R}^{n \times m}.$$

Indeed, for any $f \in \mathcal{F}$, we know

$$\max_{i \in [n]} \max_{\delta \in \mathcal{C}_B(\frac{\epsilon}{2\lambda_1})} \left| \ell\big(\{f(A(x_i, \delta)^T (f(x_i^+) - f(x_{i,k}^-)))\}_{k=1}^K\big) - \ell(\{c_r^{ik}(\delta)\}_{k=1}^K) \right|$$

$$= \max_{i \in [n]} \max_{\delta \in \mathcal{C}_B(\frac{\epsilon}{2\lambda_1})} \left| \ell(\{h_f(x_i, x_i^+, x_{ik}^-, \delta)\}_{k=1}^K) - \ell(\{c_r^{ik}(\delta)\}_{k=1}^K) \right|$$

$$\leq \lambda_2 \max_{i \in [n]} \max_{\delta \in \mathcal{C}_B(\frac{\epsilon}{2\lambda_1})} \left| \max_{k \in [K]} \ell(h_f(x_i, x_i^+, x_{ik}^-, \delta)) - \max_{k \in [K]} \ell(c_r^{ik}(\delta)) \right|$$

$$\leq \lambda_2 \max_{i \in [n]} \max_{\delta \in \mathcal{C}_B(\frac{\epsilon}{2\lambda_1})} \max_{k \in [K]} \left| \ell(h_f(x_i, x_i^+, x_{ik}^-, \delta)) - \ell(c_r^{ik}(\delta)) \right|$$

$$\leq \lambda_2 \max_{i \in [n]} \max_{\delta \in \mathcal{C}_B(\frac{\epsilon}{2\lambda_1})} \max_{k \in [K]} \left| h_f(x_i, x_i^+, x_{ik}^-, \delta) - c_r^{ik}(\delta) \right| \leq \lambda_2 \frac{\epsilon}{2\lambda_2} = \frac{\epsilon}{2}.$$

The first inequality derives from the fact that $\ell$ is Lipschitz with constant $\lambda_2$ with respect to the $\ell_\infty$-norm. The second inequality comes from $|\max_x f(x) - \max_x g(x)| \leq \max_x |f(x) - g(x)|$ and the third inequality follows from the $\lambda_2$-Lipschitzness of the loss function $\ell$. Since the cardinality of $\mathcal{C}_{\mathcal{H}}$ and $\mathcal{C}_{\tilde{\mathcal{G}}_{adv}}$ are the same, we have $\mathcal{N}_\infty(\frac{\epsilon}{2}, \tilde{\mathcal{G}}_{adv}, \tilde{S}) \leq \mathcal{N}_\infty(\frac{\epsilon}{2\lambda_2}, \mathcal{H}, S_{\mathcal{H}})$. $\qquad \square$

Now, we are going to prove Lemma 4.2.

*Proof of Lemma 4.2.* Our goal is to control the $\ell_\infty$-covering number of the function class $\mathcal{H}$ using the covering number of the representation function class $\mathcal{F}$. We will prove the lemma in two parts.
First, we claim to show:

$$\mathcal{N}_\infty\left(\frac{\epsilon}{2\lambda_2}, \mathcal{H}, S_\mathcal{H}\right) \le \mathcal{N}_\infty\left(\frac{\epsilon}{8R\lambda_2}, \mathcal{F}, S_\mathcal{F}\right),$$

where we introduce $S_\mathcal{F}$ as follows:

$$S_\mathcal{F} = \{\tilde{x}_j : j \in [nm+nK+n]\} = \{A(x_i, \tilde{\delta}) : i \in [n], \tilde{\delta} \in \mathcal{C}_\mathcal{B}(\epsilon/(2\lambda_1))\} \cup \{x_{ik}^- : i \in [n], k \in [K]\} \cup \{x_i^+ : i \in [n]\}$$

and $m = |\mathcal{C}_\mathcal{B}(\epsilon/2\lambda_1)|$. Consider the following class of functions defined on $S_\mathcal{F}$

$$\mathcal{F}_{S_\mathcal{F}} := \left\{\left(f(\tilde{x}_1), \ldots, f(\tilde{x}_{nm+nK+n})\right) : f \in \mathcal{F}\right\} \subset \mathbb{R}^{nm+nK+n},$$

which can be expanded as:

$$\{(f(A(x_1, \delta_1)), \ldots, f(A(x_n, \delta_m)), f(x_1^+), \ldots, f(x_n^+), f(x_{11}^-), \ldots, f(x_{1K}^-), f(x_{n1}^-), \ldots, f(x_{nK}^-))\}.$$

Suppose the function class $\mathcal{F}_{S_\mathcal{F}}$ has a proper $(\epsilon/(8R\lambda_2), \ell_\infty)$-cover as below

$$\mathcal{C}_\mathcal{F} := \left\{(\tilde{c}_{i'}^1(\tilde{\delta}_1), \ldots, \tilde{c}_{i'}^n(\tilde{\delta}_m), \tilde{c}_{i'}^{1+}, \ldots, \tilde{c}_{i'}^{n+}, \tilde{c}_{i'}^{11-}, \ldots, \tilde{c}_{i'}^{nK-}) : i' \in [M]\right\} \subset \mathbb{R}^{nm+nK+n}.$$

Then for all $f \in \mathcal{F}$, there exists an $r \in [M]$ such that:

$$\max_{i \in [n]} \max_{a \in [m]} \left|f(A(x_i, \delta_a)) - \tilde{c}_r^i(\tilde{\delta}_a)\right| \le \frac{\epsilon}{8R\lambda_2},$$

$$\max_{i \in [n]} \left|f(x_i^+) - \tilde{c}_r^{i+}\right| \le \frac{\epsilon}{8R\lambda_2},$$

$$\max_{i \in [n]} \max_{k \in [K]} \left|f(x_{ik}^-) - \tilde{c}_r^{ik-}\right| \le \frac{\epsilon}{8R\lambda_2}.$$

Now, for the following function class

$$\mathcal{H}_{S_\mathcal{H}} := \left\{(f(A(x_1, \delta_1))^T(f(x_1^+) - f(x_{11}^-)), \ldots, f(A(x_n, \delta_m))^T(f(x_n^+) - f(x_{nK}^-)))\right\} \subset \mathbb{R}^{nKm}$$

projected onto the dataset $S_\mathcal{F}$, we construct a cover as follows

$$\mathcal{C}_\mathcal{H} := \left\{(c_{i'}^1(\tilde{\delta}_1), \ldots, c_{i'}^{nK}(\tilde{\delta}_m)) : i' \in [M]\right\} \subset \mathbb{R}^{nKm},$$

where $c_{i'}^{ik}(\tilde{\delta}_a) = \tilde{c}_{i'}^{i}(\tilde{\delta}_a)^T(\tilde{c}_{i'}^{i+} - \tilde{c}_{i'}^{ik-})$. We then have

$$\max_{i \in [n]} \max_{k \in [K]} \max_{a \in [m]} \left| f(A(x_i, \tilde{\delta}_a))^T(f(x_i^+) - f(x_{ik}^-)) - c_r^{ik}(\tilde{\delta}_a) \right|$$

$$= \max_{i \in [n]} \max_{k \in [K]} \max_{a \in [m]} \left| f(A(x_i, \tilde{\delta}_a))^T(f(x_i^+) - f(x_{ik}^-)) - \tilde{c}_r^{iT}(\tilde{\delta}_a)(\tilde{c}_r^{i+} - \tilde{c}_r^{ik-}) \right|$$

$$= \max_{i \in [n]} \max_{k \in [K]} \max_{a \in [m]} \left| f(A(x_i, \tilde{\delta}_a))^T(f(x_i^+) - f(x_{ik}^-)) - f(A(x_i, \tilde{\delta}_a))^T(\tilde{c}_r^{i+} - \tilde{c}_r^{ik-}) \right.$$
$$\left. + f(A(x_i, \tilde{\delta}_a))^T(\tilde{c}_r^{i+} - \tilde{c}_r^{ik-}) - \tilde{c}_r^{iT}(\tilde{\delta}_a)(\tilde{c}_r^{i+} - \tilde{c}_r^{i-}) \right|$$

$$\leq \max_{i \in [n]} \max_{k \in [K]} \max_{a \in [m]} \left( \left| f(A(x_i, \tilde{\delta}_a))^T(f(x_i^+) - f(x_{ik}^-)) - f(A(x_i, \tilde{\delta}_a))^T(\tilde{c}_r^{i+} - \tilde{c}_r^{ik-}) \right| \right.$$
$$\left. + \left| f(A(x_i, \tilde{\delta}_a))^T(\tilde{c}_r^{i+} - \tilde{c}_r^{ik-}) - \tilde{c}_r^{iT}(\tilde{\delta}_a)(\tilde{c}_r^{i+} - \tilde{c}_r^{ik-}) \right| \right)$$

$$\leq \max_{i \in [n]} \max_{k \in [K]} \max_{a \in [m]} \left| f(A(x_i, \tilde{\delta}_a))^T(f(x_i^+) - \tilde{c}_r^{i+} - f(x_{ik}^-) + \tilde{c}_r^{ik-}) \right|$$
$$+ \max_{i \in [n]} \max_{k \in [K]} \max_{a \in [m]} \left| (\tilde{c}_r^{i+} - \tilde{c}_r^{ik-})^T(f(A(x_i, \tilde{\delta}_a)) - \tilde{c}_r^{i}(\tilde{\delta}_a)) \right|$$

$$\leq \max_{i \in [n]} \max_{a \in [m]} \|f(A(x_i, \tilde{\delta}_a))\|_1 \max_{i \in [n]} \max_{k \in [K]} \|f(x_i^+) - \tilde{c}_r^{i+} - f(x_{ik}^-) + \tilde{c}_r^{ik-}\|_\infty$$
$$+ \max_{i \in [n]} \max_{k \in [K]} \|\tilde{c}_r^{i+} - \tilde{c}_r^{ik-}\|_1 \max_{i \in [n]} \max_{a \in [m]} \|f(A(x_i, \tilde{\delta}_a)) - \tilde{c}_r^{i}(\tilde{\delta}_a)\|_\infty$$

$$\leq 2R\frac{\epsilon}{8R\lambda_2} + 2R\frac{\epsilon}{8R\lambda_2} = 4R\frac{\epsilon}{8R\lambda_2} = \frac{\epsilon}{2\lambda_2}.$$

In the second equality, we added and subtracted $f(A(x_i, \tilde{\delta}_a))^T(\tilde{c}_r^{i+} - \tilde{c}_r^{ik-})$ and used the subadditivity property of absolute values in the first inequality (i.e. $|x + y| \leq |x| + |y|$). Here, the third inequality uses the property that $|x^T y| \leq \|x\|_1 \|y\|_\infty$ and the fact that $\|f(A(x_i, \tilde{\delta}_a))\|_1 \leq R$. Since the cardinality of $\mathcal{C}_\mathcal{F}$ and $\mathcal{C}_\mathcal{H}$ are the same, we have: $\mathcal{N}_\infty(\frac{\epsilon}{2\lambda_2}, \mathcal{H}, S_\mathcal{H}) \leq \mathcal{N}_\infty(\frac{\epsilon}{8R\lambda_2}, \mathcal{F}, S_\mathcal{F})$.

For the second part of the proof, we need to show

$$\mathcal{N}_\infty\Big(\frac{\epsilon}{8R\lambda_2}, \mathcal{F}, S_\mathcal{F}\Big) \leq \mathcal{N}_\infty\Big(\frac{\epsilon}{8R\lambda_2}, \tilde{\mathcal{F}}, S_{\tilde{\mathcal{F}}}\Big).$$

We introduce $S_{\tilde{\mathcal{F}}}$:

$$S_{\tilde{\mathcal{F}}} = \{(\tilde{x}, l) : l \in [ndm + ndK + nd]\}$$
$$= \{(A(x_i, \tilde{\delta}), j) : i \in [n], j \in [d], \tilde{\delta} \in \mathcal{C}_\mathcal{B}(\frac{\epsilon}{2\lambda_1})\} \cup \{(x_{ik}^-, j) : i \in [n], k \in [K], j \in [d]\} \cup \{(x_i^+, j) : i \in [n], j \in [d]\}.$$

Assume

$$\tilde{\mathcal{F}} = \{(x, j) \mapsto f_j(x) : f \in \mathcal{F}, x \in \mathcal{X}, j \in [d]\}$$

over $S_{\tilde{\mathcal{F}}}$ has a $(\epsilon/(8R\lambda_2), \ell_\infty)$-cover defined as below:

$$\mathcal{C}_{\tilde{\mathcal{F}}} := \left\{ (\tilde{c}_{i'}^{11}(\tilde{\delta}_1), \ldots, \tilde{c}_{i'}^{nd}(\tilde{\delta}_m), \tilde{c}_{i'}^{11+}, \ldots, \tilde{c}_{i'}^{nd+}, \tilde{c}_{i'}^{111-}, \ldots, \tilde{c}_{i'}^{nKd-}) : i' \in [M] \right\} \subset \mathbb{R}^{ndm + ndK + nd}.$$

This means the projection of $\tilde{\mathcal{F}}$ on the extended dataset $S_{\tilde{\mathcal{F}}}$ is:

$$\tilde{\mathcal{F}}_{S_{\tilde{\mathcal{F}}}} = \{(f_1(A(x_1, \tilde{\delta}_1)), \ldots, f_d(A(x_n, \tilde{\delta}_m)), f_1(x_1^+), \ldots, f_d(x_n^+), f_1(x_{11}^-), \ldots, f_d(x_{nK}^-))\} \subset \mathbb{R}^{ndm + ndK + nd}.$$

Then for all $f \in \tilde{\mathcal{F}}$, there exists an $r \in [M]$, such that:

$$\max_{i \in [n]} \max_{a \in [m]} \max_{j \in [d]} \left| f_j(A(x_i, \delta_a)) - \tilde{c}_r^{ij}(\tilde{\delta}_a) \right| \leq \frac{\epsilon}{8R\lambda_2},$$

$$\max_{i \in [n]} \max_{j \in [d]} \left| f_j(x_i^+) - \tilde{c}_r^{ij+} \right| \leq \frac{\epsilon}{8R\lambda_2},$$

$$\max_{i \in [n]} \max_{k \in [K]} \max_{j \in [d]} \left| f_j(x_{ik}^-) - \tilde{c}_r^{ikj-} \right| \leq \frac{\epsilon}{8R\lambda_2}.$$

Now, for the function class

$$\mathcal{F}_{S_{\mathcal{F}}} := \{(f(A(x_1, \delta_1)), \ldots, f(A(x_n, \delta_m)), f(x_1^+), \ldots, f(x_n^+), f(x_{11}^-), \ldots, f(x_{1K}^-), f(x_{n1}^-), \ldots, f(x_{nK}^-))\},$$

we construct a cover as follows

$$\mathcal{C}_{\mathcal{F}} := \left\{ (c_{i'}^1(\tilde{\delta}_1), \ldots, c_{i'}^n(\tilde{\delta}_1), c_{i'}^{1+}, \ldots, c_{i'}^{n+}, c_{i'}^{11-}, \ldots, c_{i'}^{nK-}) : i' \in [M] \right\} \subset \mathbb{R}^{nm+nK+n},$$

where $c_{i'}^i(\tilde{\delta}_a) = (\tilde{c}_{i'}^{i1}(\tilde{\delta}_a), \ldots, \tilde{c}_{i'}^{id}(\tilde{\delta}_a))^T$, $c_{i'}^{i+} = (\tilde{c}_{i'}^{i1+}, \ldots, \tilde{c}_{i'}^{id+})^T$, and $c_{i'}^{ik-} = (\tilde{c}_{i'}^{ik1-}, \ldots, \tilde{c}_{i'}^{ikd-})^T$. Therefore, we have

$$\max_{i \in [n]} \max_{a \in [m]} \|f(A(x_i, \tilde{\delta}_a)) - c_r^i(\tilde{\delta}_a)\|_\infty = \max_{i \in [n]} \max_{a \in [m]} \left| \max_{j \in [d]} f_j(A(x_i, \tilde{\delta}_a)) - \max_{j \in [d]} \tilde{c}_r^{ij}(\tilde{\delta}_a) \right|$$
$$\leq \max_{i \in [n]} \max_{a \in [m]} \max_{j \in [d]} \left| f_j(A(x_i, \tilde{\delta}_a)) - \tilde{c}_r^{ij} \right| \leq \frac{\epsilon}{8R\lambda_2},$$

$$\max_{i \in [n]} \|f(x_i^+) - c_r^{i+}\|_\infty = \max_{i \in [n]} \left| \max_{j \in [d]} f_j(x_i^+) - \max_{j \in [d]} \tilde{c}_r^{ij+} \right| \leq \max_{i \in [n]} \max_{j \in [d]} \left| f_j(x_i^+) - \tilde{c}_r^{ij} \right| \leq \frac{\epsilon}{8R\lambda_2}$$

and

$$\max_{i \in [n]} \max_{k \in [K]} \|f(x_{ik}^-) - c_r^{ik-}\|_\infty = \max_{i \in [n]} \max_{k \in [K]} \left| \max_{j \in [d]} f_j(x_{ik}^-) - \max_{j \in [d]} \tilde{c}_r^{ikj} \right|$$
$$\leq \max_{i \in [n]} \max_{k \in [K]} \max_{j \in [d]} \left| f_j(x_{ik}^-) - \tilde{c}_r^{ikj} \right| \leq \frac{\epsilon}{8R\lambda_2}.$$

The second inequality comes from $|\max_x f(x) - \max_x g(x)| \leq \max_x |f(x) - g(x)|$ and the cardinality of $\mathcal{C}_{\tilde{\mathcal{F}}}$ and $\mathcal{C}_{\mathcal{F}}$ are the same. It then follows that $\mathcal{C}_{\mathcal{F}}$ is an $(\epsilon/(8R\lambda_2), \ell_\infty)$-cover to $\mathcal{F}$. Thus, we have

$$\mathcal{N}_\infty\left(\frac{\epsilon}{8R\lambda_2}, \mathcal{F}, S_{\mathcal{F}}\right) \leq \mathcal{N}_\infty\left(\frac{\epsilon}{8R\lambda_2}, \tilde{\mathcal{F}}, S_{\tilde{\mathcal{F}}}\right).$$

The proof is completed. $\qquad\square$

## A.2 Proof of Theorem 4.1

To prove Theorem 4.1, we define the Rademacher complexity and review the theorem A.1 from Mohri et al. (2018), which controls generalization of learning algorithms by Rademacher complexity of function classes.

**Definition A.1** (Rademacher complexity). Given a class of real-valued functions $\mathcal{F}$ and dataset $S = \{z_i\}_{i=1}^m$ drawn from the distribution $\mathcal{D}$ over a space $\mathcal{Z}$, the empirical Rademacher complexity of $\mathcal{F}$ w.r.t. $S$ is defined as $\mathfrak{R}_S = \mathbb{E}_\epsilon[\sup_{f \in \mathcal{F}} \frac{1}{m}\Sigma_{i \in [m]}\epsilon_i f(z_i)]$, where each $\epsilon_i$ is an independent Rademacher variable, uniformly distributed over $\{+1, -1\}^m$. The worse-case Rademacher complexity is then $\mathfrak{R}_{\mathcal{Z},m} = \sup_{S \subset \mathcal{Z}:|S|=m}[\mathfrak{R}_S(\mathcal{F})]$.

**Theorem A.1** (Mohri et al. 2018). Let $S = \{z_i\}_{i=1}^m$ be i.i.d. random sample from a distribution $\mathcal{D}$ defined over $\mathcal{Z}$. Further let $\mathcal{F} \subset [0,1]^{\mathcal{Z}}$ be a loss class. Then for all $\delta \in (0,1)$, we have with probability at least $1 - \delta$ over the draw of the sample $S$, for all $f \in \mathcal{F}$ that

$$R(f) \leq \hat{R}(f) + 2\mathfrak{R}_S(\mathcal{F}) + 3\sqrt{\frac{\log(2/\delta)}{2n}}.$$

Our approach relies, however, on another complexity measure, namely $\ell_\infty$-covering numbers. The following classical result of Dudley's entropy integral (Boucheron et al., 2003; Bartlett et al., 2017; Ledent et al., 2021a; Srebro et al., 2010) gives a relationship between the Rademacher complexity and $\ell_\infty$-covering number. We apply the version by Srebro et al. (2010).

**Theorem A.2** (Srebro et al. 2010). Let $\mathcal{F}$ be a class of functions mapping from a space $\mathcal{Z}$ and taking values in $[0, b]$, and assume that $0 \in \mathcal{F}$. Let $S$ be a finite sample of size $m$ and $\hat{\mathbb{E}}[f(z)^2] := \frac{1}{m}\sum_{i=1}^m f(z_i)^2$. Then

$$\mathfrak{R}(\mathcal{F}) \leq \inf_{\alpha > 0} \left( 4\alpha + \frac{12}{\sqrt{n}} \int_\alpha^{\sup_{f \in \mathcal{F}} \sqrt{\hat{\mathbb{E}}[f(z)^2]}} \sqrt{\log \mathcal{N}_2(\epsilon, \mathcal{F}, S)} d\epsilon \right).$$

We are now ready to present the proof of Theorem 4.1.

*Proof of Theorem 4.1.* The proof is a direct application of Theorems A.2 and A.1. With probability at least $1 - \delta$, for all $f \in \mathcal{F}$ and $\delta \in (0, 1)$, we have,

$$
\begin{aligned}
L_{\mathrm{un}}^{\mathrm{adv}}(f) &\leq \hat{L}_{\mathrm{un}}^{\mathrm{adv}}(f) + 2\mathfrak{R}_S(\mathcal{G}_{\mathrm{adv}}) + 3B\sqrt{\frac{\log(2/\delta)}{2n}} \\
&\leq \hat{L}_{\mathrm{un}}^{\mathrm{adv}}(f) + \inf_{\alpha > 0}\left(8\alpha + \frac{24}{\sqrt{n}}\int_\alpha^B \sqrt{\log\mathcal{N}_\infty(\epsilon, \mathcal{G}_{\mathrm{adv}}, S)}d\epsilon\right) + 3B\sqrt{\frac{\log(2/\delta)}{2n}} \\
&\leq \hat{L}_{\mathrm{un}}^{\mathrm{adv}}(f) + \inf_{\alpha > 0}\left(8\alpha + \frac{24}{\sqrt{n}}\int_\alpha^B \sqrt{\log\mathcal{N}_\infty(\frac{\epsilon}{8R\lambda_2}, \tilde{\mathcal{F}}, S_{\tilde{\mathcal{F}}})}d\epsilon\right) + 3B\sqrt{\frac{\log(2/\delta)}{2n}}.
\end{aligned}
$$

The first and the second inequality follow from Theorem A.1 and Theorem A.2, respectively. The final inequality is derived from Lemmas 4.1 and 4.2. $\qquad\square$

