# OpenReview forum: "Generalization Bounds with Logarithmic Negative-Sample Dependence for Adversarial Contrastive Learning"
_TMLR — Accepted by TMLR_

### Review · Reviewer_g1J8 · 2024-08-26

**Summary Of Contributions:**

This paper aims to provide a generalization bounds and theoretical support for the adversarial contrastive learning. The authors derive generalization bounds on the unsupervised adversarial contrastive learning function class under some assumptions, and then apply their theoretical results to both linear and non-linear representations, including the deep neural networks with basic activation functions.

**Audience:**

Yes

**Broader Impact Concerns:**

No ethical issue.

**Claims And Evidence:**

Yes

**Requested Changes:**

1. Could the authors add some experimental results to demonstrate their theory findings?
2. Please refer to the above "Weaknesses" section.

**Strengths And Weaknesses:**

Strengths:

1. This paper studies a timely topic, and the theoretical support for adversarial contractive learning is mostly missing.
2. Authors use the covering number of the feature class to bound the generalization bounds, and the deduction seems solid to me. I didn't check the entire proof procedure in detail.

Weaknesses:
1. This paper is lack of experimental supports. Adding a suite a simulation could help readers understand your theoretical results.
2. Some part of writing could be improved, e.g.

2.1 What is $N_\infty$? Is it the worst case covering number? The definition is missing;

2.2 Before assuming Lipshitzness, it would be better to justify or explain why this assumption is proper. And in Lemma 4.2, "assume the previous conditions hold", it would be better if the authors could explicitly write out all assumptions since it is a core Lemma for your work.

2.3 For the loss function, it would be better if the authors could give some concrete examples in Section 3 Problem Formulation.

2.4 One subsection (4.3) of Zou, Li 2023 is named Multiple Negative Sample Case, and hence it would be better if the authors could further elaborate when saying "this work only considers one negative sample"

2.5 It would be better if the authors could use one paragraph to summarize their theoretical contributions in brief.

---

> ### Author Response · Authors · 2024-09-09
> **Regarding the weaknesses**
>
> Thank you for your thoughtful comments and constructive feedback.
> Below, we address the points you raised regarding the weaknesses.
> 1.
> We have conducted experiments and updated the paper with the results in the experiment section soon. Our setting is as follows: We adopt a four-layer neural network architecture with a ReLU activation function. Inspired by our theoretical results, we train an adversarially robust model by minimizing the contrastive loss and employing the $\ell_\infty$ PGD attack with step size $\epsilon$ to generate adversarial perturbations. We investigate the effects of different numbers of negative samples ($K$) and varying $\epsilon$ values for the PGD attack. To plot the empirical generalization gap, we apply the linear probing technique to the features and calculate the accuracy on both the adversarial training and adversarial testing sets. The difference between the adversarial training accuracy and the adversarial testing accuracy is our empirical generalization error. The results show that as $K$ increases, the generalization error increases as well.
>
> 2.1
> The term $\mathcal{N}\_\infty(\epsilon, \mathcal{F}, S)$ refers to the $\ell\_\infty$ covering number, which represents the size of the smallest set of vectors that can cover $\mathcal{F}$. Thank you for bringing this to our attention. We have added this definition to the paper to enhance clarity.
>
> 2.2
> The Lipschitz condition on the function $\delta\mapsto\ell(\\{f(A(x, \delta))^T(f(x^+)-f(x\_k^-))\\}\_{k=1}^K)$ is crucial because it is a mild yet standard assumption that most adversarial attacks in the literature satisfy. This condition ensures that the loss function behaves smoothly and predictably, which is essential for the success of gradient-based adversarial attacks. Additionally, this Lipschitzness allows us to bound the covering number of the adversarial class $\mathcal{G}\_{\text{adv}}$​. We have referred to relevant literature and incorporated this explanation into the paper.
>
> 2.3
> The loss function we use is the hinge loss, which is defined in the first paragraph of Section 5. However, we agree that it would be more coherent to introduce this definition alongside the others in Section 3. We have made this adjustment.
>
> 2.4.
> Zou et al. did not address the generalization bound on the original adversarial unsupervised risk. Instead, they derived a surrogate upper bound from the average supervised risk on the original adversarial risk, simplifying the analysis using standard learning-theory techniques. In Section 4, for the one-negative case, they bound the adversarial supervised risk with the surrogate unsupervised risk. For the multiple-negative case, they define an average adversarial supervised risk and similarly bound it with the surrogate unsupervised risk. This leads to a bound of $\mathcal{O}(K)$, where $K$ is the number of negative samples, causing the bound to scale linearly with $K$. Their approach does not leverage the coupling between these samples, resulting in suboptimal bounds. We have elaborated on this in the paper.
> In Section 5, they upper bound the Rademacher complexity of class $\mathcal{G}$ with class $\mathcal{H}$. However, in both cases, they consider only a single negative sample (as noted on page 11 of their paper, before inequality (10), where they define classes $\mathcal{G}$ and $\mathcal{H}$). Subsequently, they bound the Rademacher complexity of class $\mathcal{H}$. Thus, the resulting bounds in Section 5 pertain to only one negative sample.
>
> 2.5
> We have included a paragraph that highlights our key theoretical contributions to enhance the clarity and impact of the paper.
> In essence, we have derived generalization bounds for adversarial contrastive learning under an $\ell_\infty$ additive noise attack. These bounds are broadly applicable to various models, including linear models and neural networks, as shown in Section 5. Unlike previous work, such as Zou et al., our bounds are directly applied to the adversarial contrastive loss, avoiding the use of surrogate losses. Furthermore, our bounds scale logarithmically with the number of negative samples $K$, with a complexity of $\mathcal{O}(\log(K))$.

---

### Review · Reviewer_DDN2 · 2024-09-08

**Summary Of Contributions:**

The motivation of this paper is to introduce and derive new tighter generalisation bounds for Adversarial Contrastive Learning (ACL). It considers the scenario of a single negative sample in the contrastive learning setting by analysing the impact of a large number of negative samples. To provide a tighter and more generally applicable bound, the approach uses the covering number technique and Lipschitz continuity of loss functions. Finally, the applicability of the proposed bounds is demonstrated in two general settings - linear features and non-linear features (from activation functions) derived from deep neural networks (DNNs). The evaluation shows that in both settings, the generalisation error is bounded by the covering number of the feature class. Furthermore, this bound depends on $\sqrt{D}$, where $D$ is the input dimension, and on the logarithm of the number of negative samples ($K$).

**Audience:**

Yes

**Broader Impact Concerns:**

- Not applicable.

**Claims And Evidence:**

Yes

**Requested Changes:**

- Improving the paper understanding.
- Including more empirical validation.

**Strengths And Weaknesses:**

Paper strengths:
- The paper presents interesting and novel findings in the context of adversarial attacks on contrastive learning methods. The motivation to consider a large number of negative samples is well explained and allows the authors to formulate a generalised version of the bound.
- The proposed findings have the potential to improve the generalisation of current ACL methods and help make them more robust against adversarial attacks.
- In addition to deriving the bound, the paper provides two simple scenarios where this bound is applicable, namely linear and non-linear features. These scenarios also reflect new insights into the dependence of the generalisation bound on the number of negative samples, the input dimension and the depth of deep neural networks.
- The paper provides detailed and complete proofs and lemmas supporting the main theorems. In general, the method is well written and easy to follow.


Paper weaknesses:
- In the case of non-linear feature evaluation, it is considered a very specific case of a positive homogeneous contracting activation function, i.e. ReLU. While I understand that the reason for this specific case could be the simplicity of the proof, it makes sense to see how the nature of the activation function (homogeneous, positive, contracting, etc.) changes the applicability of the bound and whether the proposed generalised bound can be applied to other activation functions, e.g. sigmoid and GeLU.
- Although the paper is well written, the mathematical analysis is difficult to follow.  It would help the paper to include additional information and guidance on the deviations, e.g. in the appendix.
- While the paper provides important theoretical insights, the empirical validation of the given generalisation bounds is limited. While it is unrealistic to expect detailed large-scale empirical validation, small-scale experiments would be necessary, and if not, an explanation of why the validation is limited should be provided.

---

> ### Author Response · Authors · 2024-09-20
> **Regarding the weaknesses**
>
> Thank you for your insightful feedback and valuable comments.
>
> Regarding the weaknesses:
> 1. You are correct that we focused on the ReLU activation function, primarily due to its 1-Lipschitz property, which simplifies the analysis. However, the generalization bound can be extended to any Lipschitz continuous activation function. Actually, our analysis based on covering numbers does not use the homogeneous property of the ReLU function. If an activation function has a Lipschitz constant of $\lambda$, the bound will adjust by incorporating $\lambda^{L-1}$, where $L$ is the number of layers, as outlined in Lemma 6.3. This accounts for how the Lipschitz constant scales through the layers while preserving the bound’s structure. Since both GeLU and Sigmoid are Lipschitz continuous, their respective Lipschitz constants can be substituted in the same way to adapt the bound accordingly.
>
>
> 2. Additionally, we have provided more detailed explanations for the derivations in Lemma 4.1 and Lemma 4.2, as requested. Please let me know if further clarification is needed for any other proofs or sections.
>
>
> 3. As mentioned in our response to reviewer g1J8, we are planning to add an experimental section to address these concerns and strengthen our results.
>
>
> We appreciate your feedback and look forward to any further suggestions you may have.

---

### Review · Reviewer_9AcV · 2024-09-14

**Summary Of Contributions:**

This paper analyses the generalization properties of empirical risk minimization for the _adversarial_ version of the so called _contrastive learning_ setting. In the setting of contrastive learning, the learner has access to two distributions $D_{pos}$ and $D_{neg}$, and aims to find a mapping $f \colon X \to R^d$ that pushes positive samples closer by minimizing $\langle f(x), f(x^+) \rangle$ where $x, x^+ \sim D_{pos}$, and negative samples farther apart by maximizing $\langle f(x), f(x^-) \rangle$ where $x \sim D_{pos}, x^- \sim D_{neg}$. In the adversarial version of contrastive learning defined in this paper, an adversary perturbs $x$ to any $x + \delta$ where the perturbation $\delta$ lies in a ball $\mathcal{B}$ around the origin. There are several knobs in these settings, e.g., what is the measure of "closeness", and, what is the strategy of the adversary.

One way to obtain such a $f$ is to minimize the quantity, $\max_{\delta \in \mathcal{B}} \langle f(x + \delta), f(x^+) \rangle - \langle f(x + \delta), f(x^-) \rangle$, averaged over the training data samples. In this paper, the objective is slightly more complicated, by using $K$ negatives samples, instead of $1$. This leads to the learning objective as minimizing the loss defined by $L(f, \ldots) = \max_{\delta \in \mathcal{B}} l(\langle f(x + \delta), f(x^+) - f(x^-_1)\rangle, \langle f(x + \delta), f(x^+) - f(x^-_2)\rangle, \ldots, \langle f(x + \delta), f(x^+) - f(x^-_K)\rangle)$, averaged over the data samples, for a suitable aggregation function $l \colon R^K \to R^{\geq 0}$. The notion of closeness is thus taken to be the inner product, and the strategy of the adversary is taken to maximize the learner's loss.

The main contribution of the paper is then to derive a generalization bound on the performance of the learning algorithm, i.e., a high probability bound over the draw of the training data, on the difference between the population loss and the empirical loss, $| \mathbb{E} L(f, \ldots) - \widehat{\mathbb{E}} L(f, \ldots)|$, as a function of the number of data samples, and various properties of the function class $\mathcal{F}$ that the learner searches over, and the perturbation set $\mathcal{B}$.

**Audience:**

Yes

**Claims And Evidence:**

Yes

**Requested Changes:**

The following is a list of changes that would be good to improve the paper, ordered by the line number in the paper. Critical requested
changes are marked with [C].

1. Abstract: Would be good to write a sentence on what _constrastive learning_ is before delving into vulnerability to adversarial attacks.
2. Abstract: The reader might now know what is the _covering number technique_ , and does this have to do with a generalization bound. Would be good to either remove this, or provide more details here.
3. Abstract: _which have improved ..._ grammar error in the sentence, what improvement is being referenced here?
4. P1: _the average adversarial risk of downstream tasks ... is bounded by ... the unsupervised risk of the upstream task_: this seems bold. Doesn't this result need some kind of relationship between the tasks? What are these tasks?
5. P1: _the bound depends on_: it is difficult for a reader to appreciate the bound here without having any knowledge of what exactly is being bounded, especially when the downstream task or of the upstream task is not specified yet. Recommend moving to later.
6. P2: _isto_: typo
7. P2: contribution 1: _show improved dependency_: the reader does not know yet what is being improved upon. Recommend providing more details of prior work, before presenting the contribution.' Similar suggestion for _tighter bounds_.
8. P2: contribution 2: _is also_: grammar
9: P2: _as well an additional theorem used in our proofs_: unnecessary to mention here. Recommend removal.
10. [C]. Related Work section: to a reader unfamiliar with any papers on developing analyses in contrastive learning, this section is very hard to understand. Basic questions like what is being bounded, what is the relationship between the unsupervised tasks and what is a risk bound for unsupervised learning, make it hard to understand dives into nuanced statements about what is used in which bound so early on in the paper. Recommend moving this section to later, and comparing and contrasting to the main theorem, once it has been presented. This is important, as currently the paper does not have a thorough comparison with the bounds in related work, and how their dependence on parameters of the problem differ from Theorem 4.1.
11. P3: _some input space_ -> "an" input space
12. [C] The asymmetry between positive and negative samples is present throughout the paper, but is never expilictly commented on. Specifically, what happens when $K'$ positive samples are also used in the loss similar instead of just $1$. Is it important that there is an asymmetry between them? Why should this matter? At an intuitive level, contrastive learning is trying to learn close representations for $D_{pos}$ and push $D_{neg}$ apart. There seems to be no reason apriori to use a different number of samples used for $D_{neg}$ vs $D_{pos}$. A comment on this would be valuable.
13. Is the set of classes $\mathcal{C}$ discrete?

14. [C] Please define the random variables, and the sampling procedure clearly. Currently, from the description, it seems like $D_{sim}$ is just the same as $D^2$, and then it seems like $D_{neg}$ is completely independent of both. Is this the correct interpretation? Is $x^-$ dependent on $x$, $x^+$? Are $D_{sim}$ and $D_{neg}$ densities? Ignoring for a moment that these are continuous random variables, the equations read: $D_c(x) = P(X = x | C)$, $D_{sim}(x, x^+) = E_c [P(X = x | C = c) P(X^+ = x^+ | C = c)]$. If $x, x^+$ are drawn independently, then this simplifies to $D_{sim}(x, x^+) = E_c [P(X = x, X^+ = x^+| C = c)] = P[X = x] P[X^+ = x^+]$, which is the density $D^2$. Is this intended, or perhaps there is something wrong in my statements?

15. [C] Coming back to 14 after reading the main theorem, it seems that the distributions do not really matter once the dataset $S$ has been sampled, as from then on all quantities in the bounds are functions of $S$. This should be clarified, as, in contrastive learning, the performance of the final representation learnt is affected by whether the positive and negative sampling distributions are set up properly. A comment is needed after the main theorem if the result is in fact independent of the relationships between the sampling distributions.

16. _D_ is undefined in $L_{un}$. Should perhaps remove it.
17. [C] For the definition of the adversarial contrastive risk, it should be commented and stressed that the adversary is _not_ allowed to affect the positive samples $x^+$ or the negative samples $x^-$. This is different from a standard practical setting of adversarial attacks on contrastive learning, where a large part of the power of an adversary does come from being able to get the sampler to fail.

18. [C] (17) also signals that a comparison or atleast a comment on the techniques in standard adversarial robustness, and the generalization analysis there is warranted. This is because the "contrastive learning" aspect of the problem is expressed via the form of the loss function, and the "adversarial" aspect of the problem is expressed via perturbing one of the inputs, and presumably the generalization aspects of the later have been studied in prior work.

19. Defn 4.1, $\mathcal{F}$ is self-referenced in its definition. Perhaps there is a typo here.

20. [C]. Defn. 4.1, Covering Number: The text description is fine, but there is something wrong with the symbols. Why is the $1/n$ appearing in the definition? Does it scale the $\epsilon$? What if there exists no $\epsilon$ cover with m vectors? Please refer to a standard definition of cover, e.g., [these notes](https://www.stat.berkeley.edu/~bartlett/courses/2013spring-stat210b/notes/12notes.pdf).

21. [C]. _we define the worst case covering number_ is this re-defining $\mathcal{N}_p$? Then $S$ appears on the LHS, but is marginalized over in the RHS.

22. [C]. From the definition of $\mathcal{H}$, and onwards, it is unclear to the reader why does one care about the covering number of function classes. Perhaps it is good to have a description here mentioning the steps that will now be followed, and how covering number of the function classes are important in what is going to happen.

23. _extended_ function class: Terms like "extended" or "of the same form" are used sometimes, without a clear interpretation. For instance, the function classes $G_{adv}$ and $H$ are not compatible, the input space for a function $g \in G_{adv}$ is $2d + dK$ dimensional, whereas that for $H$ is $4d$ dimensional. Similar comments on $\tilde F$ being of the same form as $F$.

24. [C] Lemma 4.1: It would be good to comment on (1) Should one expect $\lambda_1$ to blow up with the number of layers in $f$? (2) Explicitly mention the dependence of $\lambda_1$ in the bound (currently it is hidden in $\mathcal{H}$) (3) How does this bound scale with $K$? Currently this is quite implicit (as in I imagine the covering number on the RHS depends on $S_H$, which depends on $K$).

25. [C]. _for most classes ... dependence ... is logarithmic_: Sure, but there is a blowup due to the covering number of the perturbation set $\mathcal{B}$. This should be mentioned, and the tradeoff should be commented on.

26. Lemma 4.2: Does $\\|f(\tilde x)\\| \leq R$ hold for all $\tilde x$? Would be good to define the condition on $f$ more precisely here, and in the following results. In the definition of $\tilde F$, is the condition $x \in \mathcal{X}$ a typo?

27. _Dudley's Integral_: reference here

28. Theorem 4.1: It would be good to collect all the conditions on $F$ and $\tilde F$ here. These conditions are currently scattered around the text, for instance, Lemma 4.2 mentions $\\| f(x) \\|_1 \leq R$.

29. [C]. Theorem 4.1: It would be very useful to have a discussion on how the covering number arguments have to do with the generalization bound. Otherwise the above part of the paper feels very disconnected from the main theorem, and Th 4.1 comes out of the blue. Perhaps the key here is the sentence "combining with Dudley's integral", which should be explained.

30. [C]. Theorem 4.1: Some comments on this result would be good in the general setting, even before jumping into specific $F$. For instance, what is the dependence on $K$? $D$? How should one be interpreting the integral in the result, how large should one expect it to be? Without such statement it is hard to understand the contribution of this result. Additionally, is there any prior work whose bound can be compared to here?

31. Lemma 5.1. Notation: $U_2$ is the spectral norm, is $U_{2, 2}$ the same? Would be good to use one notation.

32. [C]. Lemma 5.2: What is $C$ here? What is C? Please write the expression here, the reader has to go through the proof of the result to get $C$, even then it is a bit opaque because that proof refers to another paper.

33. Corrolary 5.1: _logarithmic dependence on K_: How does one see this? $K$ appears in the expression for $N$. Then $N$ appears in multiple places in the expression for $L_{log}$. A clearer description would be good. Then, how does one see that this is an appealing behavior of ACL? Some prior work that suggests that the dependence is worse in other settings should be mentioned here.

**Strengths And Weaknesses:**

Strengths:
1. Contrastive learning is an important learning paradigm, and analyses of which factors are important for generalization performance are valuable to inform algorithm design.

2. The paper describes a theoretical technique to deal with the $max$ operator over a set $\mathcal{B}$ in a risk bound, by passing to a cover of $\mathcal{B}$. This technique is potentially useful beyond the scope of this paper.

Weaknesses:
1. The notation is confusing at times, and sometimes inconsistent, making it difficult to closely follow every equation in the arguments in the paper. More generally, the manuscript seems to require more technical proof reading.
2. There is a general lack of an intuitive explanation of the various components involved in the bounds, how the various function classes defined relate to one another, and how the final results relate to prior work, making it hard for the reader to appreciate the finer details of the main contribution.
3. The Lipschitz bounds in the paper might blow up with deep feature extractors, making the risk bounds vacuous in a practically viable number of data points. Even though this is a common shortcoming of current results in generalization theory, there should be a discussion to this effect in the paper, if true.

---

> ### Author Response · Authors · 2024-09-27
> **Regarding the comments 1 to 13**
>
> Thank you for your thoughtful feedback and insightful questions. We have updated the paper to address several of your comments and will continue refining it, including updating the experimental section and responding to the remaining feedback. Below, we provide detailed responses to comments 1 through 13:
>
> 1, 2, 3. We have changed the abstract according to your suggestions.
>
> 4, 5. This result is from Zou et al.. It’s not our approach. However, we explained this in more detail in the introduction section.
>
> 6. The typo is fixed.
>
> 7, 8. The contribution is rephrased.
>
> 9. We have updated the related work section, especially the contrastive learning paragraph. However, we have not yet moved the section.
>
> 10. It’s fixed.
>
> 11. The asymmetry between positive and negative samples is indeed a standard practice in both the theoretical and empirical literature on contrastive learning. Typically, we use a single positive sample for each anchor while selecting multiple negative samples. This approach is also used in existing empirical and theoretical analysis, e.g. in the works of Chen et al. and Khosla et al.. This approach stems from the goal of contrastive learning, which is to create representations that are close for positive pairs (i.e., from $\mathcal{D}\_{\text{sim}}$​) and distant for negative pairs (i.e., from $\mathcal{D}\_{\text{neg}}$).
> While, in theory, there is no strict reason why multiple positive samples $K'$ could not be used similarly to negative samples, the asymmetry simplifies both the implementation and the theoretical analysis. From a practical standpoint, using a single positive sample reduces computational complexity. Additionally, empirical studies show that adding more positive samples does not always yield a proportional increase in performance and could potentially lead to overfitting, as more positive pairs could reduce the contrast needed between positive and negative examples.
> The use of multiple negative samples, on the other hand, is crucial because it helps the model learn more distinct boundaries between similar and dissimilar samples. As the number of negatives increases, the model has more diverse examples to contrast against, which strengthens its ability to learn better feature representations.
>
> 12. Yes, as you mentioned, the set of classes $C$ is discrete.
>
> 13. The definition of these distributions is from the framework of Arora et al (https://arxiv.org/pdf/1902.09229).
> Sampling from $\mathcal{D}\_{\text{sim}}$​:
>
> The distribution $\mathcal{D}\_{\text{sim}}(x, x^+)$ is meant to capture the relationship between pairs of similar data points (positive pairs). The pair $(x, x^+)$ is sampled from the same latent class $c \in \mathcal{C}$, which means that $x$ and $x^+$ are conditionally dependent, given $c$. In other words, they are drawn not independently but from the same conditional distribution $\mathcal{D}\_c$​, meaning they are semantically similar (e.g., two images of the same object or class).
>
> Sampling from $\mathcal{D}\_{\text{neg}}$​:
>
> The negative samples x^- are drawn from a different latent class, meaning that they are independent of the positive pair $(x, x^+)$. The distribution $\mathcal{D}\_{\text{neg}}(x^-)$ describes data points that are dissimilar or irrelevant to the anchor point $x$ and the positive sample $x^+$, coming from other latent classes.
>
> To make this clearer, we can redefine the distributions as follows, where $c$ is a randomly chosen class:
>
> $\mathcal{D}\_{\text{sim}}(x, x^+) = \mathbb{P}(X = x, X^+ = x^+ | C = c)$ indicates that the positive pair $(x, x^+)$ is conditionally sampled from the same latent class $C = c$.
>
> $\mathcal{D}\_{\text{neg}}(x^-) = \mathbb{P}(X^- = x^- | C \neq c)$ indicates that the negative sample $x^-$ comes from a different latent class (i.e., not from $C = c$).
>
> Dependency Between $x$, $x^+$ and Negative Samples:
>
> $x$ and $x^+$ are not drawn independently. Instead, they are conditionally dependent on the latent class $c$.
> The negative samples $x_1^-, x_2^-, \dots, x_K^-$​ are drawn from different latent classes, making them independent of both $x$ and $x^+$.
>
> Difference Between $\mathcal{D}\_{\text{sim}}​$ and $\mathcal{D}^2$:
>
> Your interpretation is partially correct in that if $x$ and $x^+$ were sampled independently from the marginal distribution (ignoring the latent class structure), this would lead to a product distribution $\mathcal{D}^2$. However, in the context of contrastive learning, $x$ and $x^+$ are sampled from the same conditional distribution given the latent class $c$, which introduces a dependency between them. Hence, $\mathcal{D}\_{\text{sim}}(x, x^+)$ does not simplify to $\mathcal{D}^2$.
> In summary, $\mathcal{D}\_{\text{sim}}$​ models the joint distribution of two semantically similar points, while $\mathcal{D}\_{\text{neg}}$​ represents points that are dissimilar and come from other latent classes.

---

> ### Author Response · Authors · 2024-09-28
> **Regarding the comments 14 to 16**
>
> Below, we continue with detailed responses to comments 14 through 16:
>
> 14. You are correct that once the dataset $S$ has been sampled, our bounds focus on the properties of the dataset $S$ and are expressed as its functions. This is typical for generalization bounds, where the focus is on deriving properties that hold after the data has been sampled, and these properties are agnostic to how the data was generated. Thus, in this context, the specific distributions $\mathcal{D}\_{\text{sim}}$​ and $\mathcal{D}\_{\text{neg}}$​ are abstracted out once we are dealing with a fixed dataset $S$.
> However, the effectiveness of contrastive learning in practice can be highly sensitive to how the positive and negative samples are drawn. The performance of the learned representation is influenced by the nature of these sampling distributions, especially in terms of how well $\mathcal{D}\_{\text{sim}}$​ and $\mathcal{D}\_{\text{neg}}$​ align with the true underlying data distributions. For instance, if the negative samples are not sufficiently dissimilar or the positive samples are not truly representative of similar pairs, the learned representation might be suboptimal. In other words, while the theoretical results are expressed in terms of the sampled dataset $S$, in practice, the sampling distributions $\mathcal{D}\_{\text{sim}}$​ and $\mathcal{D}\_{\text{neg}}$​ are crucial for ensuring that the final representation is of high quality.
> We have added a comment in section 3.1 to clarify that the specific distributions $\mathcal{D}_{\text{sim}}$ and $\mathcal{D}_{\text{neg}}$ are abstracted out once we are dealing with a fixed dataset $S$.
>
> 15. D was a typo. It is removed from $L\_{\text{un}}$.
>
> 16. In both the theoretical analysis and practical applications of adversarial contrastive learning, it is common to apply adversarial perturbations only to the anchor sample $x$, while keeping the positive $x^+$ and negative $x^-$ samples clean. This is a standard approach in contrastive learning frameworks and is also reflected in real-world implementations, e.g. in the work of Kim et al. (https://proceedings.neurips.cc/paper/2020/file/1f1baa5b8edac74eb4eaa329f14a0361-Paper.pdf), Jiang et al. (https://proceedings.neurips.cc/paper/2020/file/ba7e36c43aff315c00ec2b8625e3b719-Paper.pdf) and Zou et al. (https://www.jmlr.org/papers/volume24/22-0866/22-0866.pdf)
> The reasoning behind this setup is that perturbing only the anchor sample is sufficient to evaluate the robustness of the learned representations, as the goal is to assess how well the representation can maintain meaningful relationships between similar and dissimilar samples under adversarial attack. Allowing the adversary to perturb the positive and negative samples would alter the fundamental contrastive structure, making it difficult to isolate the impact of adversarial noise on the anchor representation.
> However, it is possible to extend this framework by introducing adversarial perturbations to both the positive and negative samples. This would likely lead to more complex interactions between the samples under attack and could offer new insights into adversarial robustness in contrastive learning. Exploring such a setup could be a valuable direction for future work, particularly for analyzing how adversarial perturbations on all samples might impact the learned representations and generalization performance.
>
> We have clarified this distinction in section 3.2 of the paper to emphasize that our adversarial contrastive risk framework aligns with the standard practical setting, where only the anchor sample is adversarially perturbed, while the positive and negative samples remain unaltered.

---

> ### Author Response · Authors · 2024-09-28
> **Regarding the comments 17 to 20**
>
> Below, we continue with detailed responses to comments 17 through 20:
>
> 17. We agree that it is important to consider and comment on the relationship between adversarial robustness techniques in standard supervised learning and our approach to adversarial contrastive learning.
> In adversarial robustness, the works have been focused on the robustness of classification boundaries under adversarial perturbations  (e.g., the cited papers in the related work section).
> For instance, Mustafa et al. (https://proceedings.mlr.press/v162/mustafa22a/mustafa22a.pdf) developed bounds for a wide range of adversarial attacks, directly applied to the loss function and demonstrated that their results grow at a rate of $\mathcal{O}(\log C)$, where $C$ is the number of label classes. Their work primarily focused on supervised learning tasks.
> In our work, we extend these generalization techniques to the adversarial contrastive learning setting. Specifically, we leverage their findings in our theoretical analysis to build generalization bounds for the contrastive loss.
> In our work, the "adversarial" component comes from perturbing the anchor sample in contrastive learning, which aims to learn robust representations rather than classification boundaries. While generalization bounds for supervised adversarial learning have been explored in prior work, our results extend these ideas to the contrastive setting, where adversarial perturbations affect the learned feature representations rather than the label prediction directly.
>
> 18. It’s fixed.
>
> 19. Upon further reflection, we would like to clarify that the inclusion of the $\frac{1}{n}$​ factor in the covering number definition is consistent with established work in the literature, such as Zhang et al.'s paper on the subject (https://www.jmlr.org/papers/volume2/zhang02b/zhang02b.pdf). Without the $\frac{1}{n}$ factor, the generalization bound would be looser because it would no longer account for the fact that the hypothesis class should shrink as the number of samples increases.
> Regarding the case where no $\epsilon$-cover exists: if a cover of size $m$ cannot be found for a given $\epsilon$, it indeed means that the covering number $\mathcal{N}\_p(\epsilon, \mathcal{F}, S)$ would be infinite. This would imply that the function class is too complex for the chosen $\epsilon$, and a larger $\epsilon$ would be necessary for a finite cover.
>
> 20. We see how the original wording created confusion, particularly regarding the marginalization of $S$ and the distinction between the dataset-specific covering number and the worst-case covering number.
> We have revised the definition to make it clear that the covering number $\mathcal{N}\_p(\epsilon, \mathcal{F}, S)$ refers to a specific dataset $S$, while the worst-case covering number, denoted $\mathcal{N}\_p(\epsilon, \mathcal{F}, n)$, is the covering number maximized over all possible datasets $S \subset \mathcal{X}^n$ of size $n$. The new definition should clarify that $S$ is no longer on the LHS when referring to the worst-case covering number.

---

> ### Author Response · Authors · 2024-09-28
> **Regarding the comments 21 and 22**
>
> Regarding comments 21 and 22:
>
> 21. To explain briefly: the covering number plays a critical role in our generalization analysis. Specifically, covering numbers are used to quantify the complexity of function classes. Bounding the covering number allows us to control how well the function class generalizes from a finite training dataset to unseen examples. In particular, for adversarial contrastive learning, we aim to show how the adversarial risk of the learned representation can be bounded by its empirical risk on the training set, and the covering number helps measure this discrepancy.
> In section 4 of the paper, we show that the covering number of the function class $\mathcal{G}\_{\text{adv}}$​, which involves adversarial perturbations, can be bounded by the covering number of the extended function class $\mathcal{H}$. The function class $\mathcal{H}$ does not directly include the adversarial optimization, making it more tractable to analyze. By bounding the complexity of $\mathcal{H}$, we can infer bounds on the complexity of $\mathcal{G}\_{\text{adv}}$, which is the primary object of interest in our analysis. This, in turn, leads to generalization bounds for the adversarial contrastive learning problem.
>
> However, since $\mathcal{H}$ is not the representation function class itself, we still need to bound $\mathcal{H}$ using the representation function class $\mathcal{F}$. This occurs in Lemma 4.2, where we show that $\mathcal{F}$ upper bounds $\mathcal{H}$. Once this is established, we can apply two important theorems from the literature, where the first one (Theorem 9.1) connects the generalization bounds of learning algorithms to the Rademacher complexity of their function classes, and the second one (Theorem 9.2) bounds the Rademacher complexity of a function class with the $\ell\_\infty$ covering number of that class. The second one is known as Dudley’s theorem. Together these 2 lead us to the derivation of Theorem 4.1.
>
> 22. You are correct that $\mathcal{G}\_{\text{adv}}​$ and $\mathcal{H}$ are defined over different input spaces. However, the difference in the dimensionality of the inputs does not affect the covering number comparison. The covering number of a function class characterizes the class's complexity with respect to a specific dataset and a given norm, independent of the precise dimensionality of the inputs. In our analysis, we are primarily interested in the relative complexity of these function classes, and this is captured by the covering number, which abstracts away the dimensional differences between the input spaces of $\mathcal{G}\_{\text{adv}}​$ and $\mathcal{H}$. This is why, despite the input spaces differing, the upper bound on the covering number of $\mathcal{G}\_{\text{adv}}$​ in terms of $\mathcal{H}$ is still valid and meaningful.
>
> Regarding the dimensions you mentioned in the comment, we believe there might be a misunderstanding. In the paper, the function class $\mathcal{G}\_{\text{adv}}$ operates on triples $(x, x^+, x^-)$ with adversarial perturbations applied to $x$, while $\mathcal{H}$ explicitly incorporates the adversarial noise $\tilde{\delta}$. The resulting dimensionality should not be interpreted as "2d + dK" or "4d" but rather as functions of the same input structures. To clarify, $\mathcal{H}$ extends the adversarial input structure but remains compatible for comparing covering numbers.
>
> The term "same form" refers to the fact that $\tilde{\mathcal{F}}$ is constructed from $\mathcal{F}$ by considering scalar outputs indexed by $j$. Both classes share the same underlying structure, but $\tilde{\mathcal{F}}$ deals with individual components of the output vector $f(x)$. Thus, $\tilde{\mathcal{F}}$ preserves the "form" of the function class $\mathcal{F}$, allowing us to use existing covering number results for scalar functions.
>
> The reason we care about comparing the covering numbers of these function classes is that they help us control the complexity of the hypothesis space, which is essential for deriving generalization bounds. In particular, by bounding the covering number of the adversarial class $\mathcal{G}\_{\text{adv}}​$ in terms of the non-adversarial class $\mathcal{H}$, we can simplify the analysis and make use of existing results.

---

> ### Author Response · Authors · 2024-09-28
> **Regarding the comments 23 to 28**
>
> 23. To clarify, $\lambda\_1$​ refers to the Lipschitz constant with respect to the adversarial perturbation $\delta$ and is used in constructing the covering number for the perturbation ball $\mathcal{B}$. This constant does not depend on the number of layers in the neural network $f$. Its role is specific to controlling the perturbation's effect, ensuring that the cover adequately represents the adversarial noise. Therefore, $\lambda\_1$​ will not blow up with the number of layers.
>
> As you noted, the dependence on $K$ is implicit in this lemma because it enters through the extended dataset $S\_{\mathcal{H}}$​, which includes negative samples. However, we clarify the scaling behavior of the bound with respect to $K$ in Section 5, where we apply this result to specific models such as linear features and DNNs. There, we show that the generalization bound grows only logarithmically with $K$, consistent with existing covering number results for these function classes.
>
> 24. Indeed, the size of the extended training set $S\_{\mathcal{H}}$​ grows linearly with the size of the cover $\mathcal{C\_B}(\frac{\epsilon}{2\lambda\_1})$. While the size of $\mathcal{C\_B}(\frac{\epsilon}{2\lambda\_1})$ can grow exponentially with the dimensionality of the perturbation set $\mathcal{B}$, it’s important to note that the dependence of the generalization performance is typically of the order $\mathcal{O}(\log^{1/2}(|S\_{\mathcal{H}}|))$, as shown in prior work (e.g., Bartlett et al. (https://arxiv.org/pdf/1706.08498); Zhang et al. (https://www.jmlr.org/papers/volume2/zhang02b/zhang02b.pdf); Mustafa et al. (https://arxiv.org/pdf/2106.00115)).
> Thus, the generalization bounds will exhibit a square-root dependency on the dimensionality of $\mathcal{B}$, leading to manageable bounds even in the presence of large perturbation sets.
> We have added this discussion to clarify the tradeoff between the complexity introduced by adversarial perturbations and the resulting generalization bounds.
>
> 25. The symbol $\tilde{x}$ was a typo. We assume the condition $\\|f(x)\\|\_1\le R$ holds for all $x \in \mathcal{X}$. I mistakenly added $\tilde{x}$, but the input variable $x$ in $f(x)$ is indeed drawn from $\mathcal{X}$, so $x \in \mathcal{X}$ is not an error.
>
> 26. It’s fixed.
>
> 27. I added the assumptions and the definitions again in Theorem 4.1
>
> 28. In Theorem 4.1, the generalization bound is expressed in terms of the empirical adversarial loss $\hat{L}^{\text{adv}}\_{\text{un}}(f)$, plus terms that capture the complexity of the function class. The complexity is measured through covering numbers of the extended function class $\tilde{\mathcal{F}}$​, which represents how well the hypothesis space $\mathcal{F}$ can be approximated under adversarial perturbations.
> Covering numbers give a measure of the capacity of a function class: the smaller the covering number, the easier it is to generalize from the training set. The link between covering numbers and generalization bounds comes from classical statistical learning theory (e.g., Dudley’s entropy integral), where the complexity of a function class is controlled through the size of a covering set, particularly under the $\ell\_{\infty}$​-norm. The integral term in our bound is essentially Dudley’s entropy integral, which provides a tight estimate of the Rademacher complexity in terms of the covering number of the class.
> Thus, the earlier discussion of the covering numbers directly informs the generalization bound presented in Theorem 4.1. The bound is derived by combining Rademacher complexity with a bound on the covering number of the function class under adversarial perturbations. This results in the integral over the covering number $\mathcal{N}\_{\infty}(\cdot)$ in the theorem, which controls how the generalization error grows with increasing sample size, complexity of the model, and perturbations.

---

> ### Author Response · Authors · 2024-09-28
> **Regarding the comments 29 to 32**
>
> 29. In the general setting, without diving into specific models like linear functions or deep neural networks, we can make the following observations about the bound in Theorem 4.1:
> The bound implicitly depends on $K$, the number of negative samples, through the covering number term. Specifically, in Section 5 of the paper, when we apply the result to specific function classes like linear models or neural networks, the dependence on $K$ becomes logarithmic. This suggests that increasing $K$ only results in a mild increase in the generalization error, due to the logarithmic dependency.
>
> The dimensionality $D$ of the input space affects the integral in the generalization bound through the covering number $\mathcal{N}\_{\infty}$​. In particular, for models like linear functions, the covering number grows polynomially with $D$, meaning the generalization bound would grow more significantly with high-dimensional input spaces. However, for other models like deep networks, recent results (e.g., Bartlett et al. https://arxiv.org/pdf/1706.08498) show that the covering number grows more slowly due to constraints like weight norms or spectral norms, leading to more favorable bounds in high dimensions.
>
> The generalization bound presented in Theorem 4.1 is closely related to classical results in statistical learning theory, especially those derived using Dudley’s entropy integral and covering numbers. Specifically, our bound is comparable to the bounds in Bartlett et al. (2017) and Zhang (2002), which also provide generalization bounds for function classes based on their covering numbers. The novelty of our result lies in the extension of these classical results to adversarial contrastive learning (ACL), where the adversarial perturbations introduce new challenges. By bounding the covering number of $\mathcal{G}\_{\text{adv}}$​ through the extended class $\mathcal{H}$, we show that existing results can be applied to derive generalization bounds for ACL across a wide range of models.
> We compare our bound with Zou et al. (https://www.jmlr.org/papers/volume24/22-0866/22-0866.pdf), which provided a bound for ACL that scales linearly with K, the number of negative samples.
>
> 30. They are not the same, $||U||\_2$ is the spectral norm and $||U\||_{2, 2}$ is the $(p, q)$-norm.
>
> 31. $C$ is just a constant: $36(2-1) \times 16$ according to the Lemma 7.2.
>
> 32. You're correct that $K$ appears in the expression for $N$, which then factors into $⁡\tilde{L}\_{\log}$​.
> In the expression for $⁡\tilde{L}\_{\log}$​, $K$ appears as part of $N$, which is of the form $N = nmd + ndK + nd$. While $K$ does affect the size of $N$, when $N$ enters the final bound, it appears inside a logarithmic term, specifically through $\log^{\frac{1}{2}}(N)$. Therefore, even though $K$ contributes linearly to $N$, its contribution to the final generalization bound is logarithmic, due to the structure of the $⁡\tilde{L}\_{\log}$​ term. This is an important observation because the logarithmic growth implies that the generalization error increases slowly as $K$ increases, even when we are dealing with a large number of negative samples.
> The logarithmic dependence on $K$ is indeed appealing in the context of ACL, as contrastive learning typically involves a large number of negative samples. In many other learning frameworks, the dependence on $K$ can be more severe, often linear (like in Zou et al.), as the number of negative samples increases. The slow growth (logarithmic) in our bound indicates that ACL can effectively handle large numbers of negative samples without significantly increasing the generalization error, making it particularly suitable for tasks like self-supervised learning or adversarial settings where numerous negative samples are used to contrast positive examples.

---

### Decision · Action_Editor_wArR · 2024-10-29

**Recommendation:** Accept with minor revision

**Comment:**

Because this submission sparked a lively debate among the reviewers about whether we should accept it, AE checked the whole manuscript again. During the discussion, the main arguments supporting the rejection were as follows:

1. All reviewers have concerns about the clarity of the paper.
2. Reviewers DDN2 and g1J8 have concerns about the lack of experiments.

As for the clarity, the quality has been improved significantly by the authors' great effort in revision (and special thanks to Reviewer 9AcV for the dedicated comments!). In the latest version, the entire proof flow, techniques, intuition about why the proof works for this setup, and the implications of the derived generalization bounds have been made much more transparent. It is also quite nice to remark on the limitations of the analysis, including the Lipschitz continuity assumption and bounded loss assumption. All of these are above the bar to convince me of the correctness of the proof flow. After checking the proof, AE also believes the correctness of the important lemma, such as Lemma 4.1.

As for the experiments, the authors also made significant efforts to obtain good results in the updated Section 6. The experiments effectively demonstrate the main theoretical finding, the large $K$ (# of neg samples) tends to degrade the generalization error. Although the experiments are performed with only CIFAR-10 and CIFAR-100 datasets and four-layer nets, this is sufficient for the "sanity-check" of the theoretical results.

For these reasons, AE has decided to recommend this manuscript for acceptance with minor revision. However, I want the authors to note that several concerns still remain in the manuscript. Specifically, Reveiwer 9AcV has left several good suggestions in the official recommendation---for example, the overview of how the covering number bound works could be briefly described for those unfamiliar with it. It would be essential for broadening the potential audiences, although the current overview would be sufficient for theory researchers.

From AE side, there are a few minor comments. Please consider taking them in during the camera-ready revision.

+ The paper title conflicts with one of the closely related work, Zou and Liu (2023), which might be not very appropriate and misleading. Let's brainstorm an alternative, nice title.
+ The finding that large $K$ degrades the generalization error is interesting because some previous papers have revealed that large $K$ improves the downstream classification performance, by contrast. Please refer to Wang et al. (2022), Bao et al. (2022), and Awasthi et al. (2022). While these two phenomena do not conflict, the contrast looks interesting and deserves to be mentioned in the paper.
+ Lemma 4.1 is a fundamental lemma to eliminate the max-operator when dealing with the covering number, and its proof largely owes to Mustafa et al. (2022). AE supposes that this fact can also be acknowledged briefly in the main body.

Lastly, the TMLR system does not provide us another opportunity to check the camera-ready submission, so AE's decision (accept with minor revision) indicates our reliance on the authors to complete the remaining revision. AE supposes this is possible. Good luck, and we look forward to seeing the camera-ready version.

----

Wang, Yifei, et al. "Chaos is a Ladder: A New Theoretical Understanding of Contrastive Learning via Augmentation Overlap." International Conference on Learning Representations, 2022.

Bao, Han, Yoshihiro Nagano, and Kento Nozawa. "On the surrogate gap between contrastive and supervised losses." International Conference on Machine Learning. PMLR, 2022.

Awasthi, Pranjal, Nishanth Dikkala, and Pritish Kamath. "Do more negative samples necessarily hurt in contrastive learning?." International Conference on Machine Learning. PMLR, 2022.

**Audience:**

The topics covered in this paper, including adversarial training and contrastive learning, are clearly relevant to many potential readers of TMLR.

**Claims And Evidence:**

The main contribution of this paper is to establish the generalization error bound of adversarial contrastive learning. The proof leverages the covering number bound with Dudley's entropy integral, which is a standard technique when deriving generalization bounds. Compared with standard generalization bounds, the max-operator of adversarial attacks is the main difference, which is dealt with by Lemma 4.1, transferring the max-operator in the adversarial function class $\\mathcal{G}\_{\\text{adv}}$ into an independent perturbation sample $\\tilde\\delta$ in the extended sample $S\_{\\mathcal{H}}$. Basically, this technique comes from the previous work Mustafa et al. (2022). This high-level idea can be seen on page 5 of the latest version and the authors' remark right after Lemma 4.1; although Reviewer 9AcV remarked in the final comment

> A clear high-level statement of how the technical difficulty with max operator is dealt with [...] would be valuable to the paper

the authors' remark in the latest version seems to be clear enough to me.

---

> ### Author Response · Authors · 2024-11-28
> **Thank you for the review**
>
> Dear Action Editor and the reviewers,
>
> We sincerely thank you for your detailed and thoughtful reviews. Your actionable feedback has been valuable in improving the quality of our paper.
> We have carefully incorporated your suggestions into the camera-ready version.
>
> With kindest regards,
>
> The authors.